# Measuring Distribution Shifts in Inverse Problems without clean data

## Abstract

Diffusion models are widely used as priors in imaging inverse problems. However, their performance often degrades under distribution shifts between the training and test-time images. Existing methods for identifying and quantifying distribution shifts typically require access to clean test images, which are never available at test time when solving inverse problems. We propose a flexible framework for measuring distribution shift using *only* corrupted test measurements and candidate diffusion model scores. Our framework enables three complementary capabilities. First, in the general case with only a pool of diffusion models, it supports a principled model selection by identifying the model whose prior best matches the test data. Second, when an in-distribution model is available, our metric provides a theoretically guaranteed estimator of KL divergence that closely matches the image-domain KL. Third, the metric serves as a tool for adaptation guidance: aligning score functions with corrupted measurements reduces the estimated shift and improves reconstruction quality. Experiments on inpainting and MRI confirm that our method (i) achieves robust model selection, (ii) reliable estimates KL divergence in the presence of an in-distribution model, and (iii) enables effective adaptation to mitigate distribution shift.

## 1 Introduction

Standard *deep learning* models typically assume that training and test data are drawn from the same distribution. However, this assumption often fails (Zhang et al., 2023), with out-of-distribution (OOD) test inputs causing significant performance degradation—specially in domains like healthcare and robotics (Yang et al., 2024). Detecting and quantifying distribution shifts is thus essential for building robust models. Recent works have focused on characterizing distribution shifts (Wiles et al., 2022; Koh & et. al, 2021; Chen et al., 2021) and detecting OOD samples (Yang et al., 2022) (see also reviews in (Salehi et al., 2022; Yang et al., 2024)). A widely used strategy for OOD detection is based on model confidence, where softmax-based indicators—such as low maximum probability or high entropy—serve as simple yet effective proxies for detecting distribution shifts, especially in classification tasks (Hendrycks & Gimpel, 2017; Liang et al., 2018).

Diffusion models (DMs) (Ho et al., 2020; Song et al., 2020) have been shown to achieve state-of-the-art performance across a wide range of tasks, including high-quality image generation (Vahdat et al., 2021; Dhariwal & Nichol, 2021; Rombach et al., 2022; Karras et al., 2022; Kim et al., 2023), imaging inverse problems (Chung et al., 2023a; 2024), and medical imaging (Chung et al., 2023b; Chung & Ye, 2022; Xie & Li, 2022; Li et al., 2024; Adib et al., 2023) (see also recent reviews (Daras et al., 2024; Kazerouni et al., 2023; Croitoru et al., 2023; Li et al., 2023)). These models approximate the score function of the data distribution and enable principled sampling via stochastic differential equations (Song et al., 2020), allowing data generation from pure noise. Since diffusion models approximate the full data distribution through learned score functions, they are inherently sensitive to distribution shifts and require efficient methods for OOD detection and shift quantification. Recent work has explored this by analyzing various diffusion model-based approaches, including score consistency, sample likelihood, reconstruction error, and properties of the diffusion trajectory (Heng et al., 2024; Graham et al., 2023b; Liu et al., 2023; Livernoche et al., 2023).

Existing methods for detecting and quantifying distribution shifts in inverse problems often assume access to clean test images, making them impractical when only corrupted measurements are observed

at test time. To overcome these limitations, we propose the first *unsupervised* framework that operates under two complementary settings: (i) when *only* corrupted measurements and candidate models are available, our method identifies the model best suited for the given test measurements; and (ii) when both in-distribution (InD) and OOD models are available, our framework estimates the KL divergence between the underlying image distributions.

Domain adaptation methods aim to mitigate distribution shifts between training and test data (Farahani et al., 2021) and are well-studied in the broader machine learning literature (Zhang & Gao, 2022; Csurka, 2017). In inverse problems, however, adaptation is particularly challenging due to the unavailability of clean test-time data. Recent *self-supervised* approaches have explored adapting deep learning models using only measurement-domain signals (Chung & Ye, 2024; Barbano et al., 2025; Darestani et al., 2022), but these methods are largely heuristic and lack a theoretical justification for their effectiveness. Our metric provides a principled framework that formally connects distribution shifts to the discrepancy between score functions, evaluated directly on corrupted measurements.This perspective not only enables model selection and KL estimation under our two settings, but also explains why aligning score functions through lightweight adaptation should reduce distribution shift. Empirically, we confirm that such adaptation lowers the estimated KL divergence and improves reconstruction quality across multiple inverse problems.

**Our contributions are: (1)** We introduce the first *unsupervised* framework for quantifying distribution shift in inverse problems using only corrupted measurements. This framework operates under two complementary settings: (i) the practical case where only a pool of diffusion models is available, enabling principled model selection for test-time data, and (ii) the special case where both InD and OOD models are available, where our metric provides a reliable estimator of the KL divergence directly from measurements. **(2)** We establish theoretical guarantees showing that, under mild assumptions on the measurement operator, the proposed measurement-domain KL divergence closely tracks the KL divergence computed from clean images. **(3)** We demonstrate that the proposed KL estimator is not only an evaluation metric but also a training objective: by aligning measurement-domain scores, it enables simple and effective adaptation of pretrained diffusion priors. Our adapted model reduce measurement-domain KL and improve reconstruction quality compared to the unadapted OOD model across inpainting and MRI tasks.

## 2 BACKGROUND

### 2.1 DENOISING DIFFUSION PROBABILISTIC MODELS

Diffusion models (Ho et al., 2020; Song et al., 2020; Karras et al., 2022) are trained to estimate the *score function* of the data distribution—that is the gradient of the log-density. During training, a forward process progressively adds Gaussian noise to clean data samples $\boldsymbol{x} \sim p(\boldsymbol{x})$ over multiple steps, while the model learns to reverse this process by denoising the corrupted samples at each step. This forward process is typically modeled as a Markov chain, $\boldsymbol{x}_{\sigma_0} \to \boldsymbol{x}_{\sigma_1} \to \cdots \to \boldsymbol{x}_{\sigma_\infty}$, where $\boldsymbol{x}_{\sigma_0} = \boldsymbol{x}$ is the clean image and noise levels $\sigma_0 < \sigma_1 < \cdots < \sigma_\infty$ increase at each step. We denote the full set of noise levels by $\boldsymbol{\sigma} := [\sigma_0, \cdots, \sigma_\infty]$, which corresponds to a time-dependent diffusion process.

The intermediate noisy variable $\boldsymbol{x}_\sigma$ is defined using a Gaussian kernel:

$$p(\boldsymbol{x}_\sigma | \boldsymbol{x}) = \mathcal{N}(\boldsymbol{x}, \sigma^2 \boldsymbol{I}),$$

which enables direct sampling via $\boldsymbol{x}_\sigma = \boldsymbol{x} + \boldsymbol{n}$, where $\boldsymbol{n} \sim \mathcal{N}(0, \sigma^2 \boldsymbol{I})$. The marginal distribution of the noisy images, denoted $p(\boldsymbol{x}_\sigma)$, is given by:

$$p(\boldsymbol{x}_\sigma) = \int p(\boldsymbol{x}_\sigma | \boldsymbol{x}) p(\boldsymbol{x}) \mathrm{d}\boldsymbol{x} = \int G_\sigma(\boldsymbol{x}_\sigma - \boldsymbol{x}) p(\boldsymbol{x}) \mathrm{d}\boldsymbol{x}, \tag{1}$$

where $G_\sigma$ denotes the Gaussian density function with standard deviation $\sigma \geq 0$.

Tweedie's formula establishes a link between Gaussian denoising and score estimation (Robbins, 1956; Miyasawa, 1961) by expressing the posterior mean in terms of the score of the noise-corrupted density:

$$\mathsf{D}_\sigma(\boldsymbol{x}_\sigma) = \mathbb{E}[\boldsymbol{x} | \boldsymbol{x}_\sigma] = \boldsymbol{x}_\sigma + \sigma^2 \nabla \log p(\boldsymbol{x}_\sigma). \tag{2}$$

This result implies that learning the Gaussian denoiser $D_\sigma$ is equivalent to learning the score $\nabla \log p(\boldsymbol{x}_\sigma)$ of the noisy distribution, for all noise levels $\sigma \geq 0$. In practice, the denoiser $D_\sigma$ is trained to minimize the mean squared error (MSE) between the clean and denoised signals:

$$\mathrm{MSE}(D_\sigma) = \mathbb{E}_{\boldsymbol{x}, \boldsymbol{x}_\sigma} \left[ \|\boldsymbol{x} - D_\sigma(\boldsymbol{x}_\sigma)\|_2^2 \right]. \tag{3}$$

A diffusion model consists of a collection of MMSE denoisers across all noise levels, $\{D_\sigma : \sigma \in \boldsymbol{\sigma}\}$, which implicitly provide access to the score functions $\nabla \log p(\boldsymbol{x}_\sigma)$ of the noise-corrupted densities. These learned score functions enable sampling from the underlying clean image distribution $p(\boldsymbol{x})$ via the reverse diffusion process (Vincent, 2011; Raphan & Simoncelli, 2011).

## 2.2 Measuring Distribution Shifts with Clean Images using Score Functions

We extend the framework introduced in (Song et al., 2021; Kadkhodaie et al., 2024) to derive an expression for the KL divergence between the InD $p(\boldsymbol{x})$ and OOD $q(\boldsymbol{x})$ densities. In particular, the KL divergence can be expressed in terms of the score functions of the corresponding noise-corrupted distributions as

$$D_{\mathrm{KL}}(p(\boldsymbol{x}) \| q(\boldsymbol{x})) = \int_0^\infty \mathbb{E}_{\boldsymbol{x} \sim p(\boldsymbol{x})} \left[ \|\nabla_{\boldsymbol{x}_\sigma} \log p(\boldsymbol{x}_\sigma) - \nabla_{\boldsymbol{x}_\sigma} \log q(\boldsymbol{x}_\sigma)\|_2^2 \right] \sigma \, \mathrm{d}\sigma. \tag{4}$$

Here, $p(\boldsymbol{x}_\sigma)$ and $q(\boldsymbol{x}_\sigma)$ denote the noise-corrupted distributions of $p(\boldsymbol{x})$ and $q(\boldsymbol{x})$, respectively, at noise level $\sigma$. The score function $\nabla_{\boldsymbol{x}_\sigma} \log p(\boldsymbol{x}_\sigma)$ can be estimated using the Tweedie's formula, which relates it to the posterior mean $\mathbb{E}[\boldsymbol{x}|\boldsymbol{x}_\sigma]$ according to Eq. (2). This posterior mean, in turn, can be approximated by training MMSE denoisers via the loss in Eq. (3).

In practice, diffusion models are trained as denoisers across a range of noise levels to approximate the score functions of the corresponding noise-corrupted data distributions. Thus, the KL divergence in Eq. (4) can be estimated when two diffusion models are available: one trained on InD samples from $p(\boldsymbol{x})$, and another on ODD samples from $q(\boldsymbol{x})$.

When using diffusion models to estimate KL divergence, it is assumed that both the InD and OOD models have accurately learned the score functions of their respective data distributions. The discrepancy between their learned Gaussian denoisers at each noise level reflects the extent of the distribution shift. Leveraging the connection between the conditional mean estimator provided by the deep MMSE denoiser and the score function from Eq. (2), we obtain a tractable metric for measuring distribution shift in image domain (see Appendix A for the proof, as well as (Song et al., 2021; Kadkhodaie et al., 2024) for additional discussion). Notably, the resulting metric corresponds to the integrated denoising gap between the InD and OOD diffusion models across all noise levels.

The KL divergence formulation in Eq. (4) quantifies the shift between the InD density $p(\boldsymbol{x})$ and the OOD density $q(\boldsymbol{x})$ only when clean InD images are available. To cover the more realistic setting in which we possess only corrupted measurements, we introduce an *unsupervised* metric that estimates the same distribution shift directly from those measurements.

# 3 Distribution Shift in Measurement Domain

Clean images required for the KL divergence in Eq. (4) are unavailable in many inverse problems. We therefore derive a measurement-domain KL estimator that quantifies distribution shift directly from the observed measurements and pretrained diffusion models.

## 3.1 Problem Formulation

We consider a set of measurement operators randomly drawn from the distribution $p(\mathbf{H})$. For a given $\mathbf{H} \in \mathbb{R}^{m \times n}$, the measurement vector $\boldsymbol{y} \in \mathbb{R}^m$ is related to the underlying signal $\boldsymbol{x} \in \mathbb{R}^n$ via

$$\boldsymbol{y} = \mathbf{H}\boldsymbol{x} + \boldsymbol{z}, \tag{5}$$

where $\boldsymbol{z} \sim \mathcal{N}(0, \sigma_{\boldsymbol{z}}^2 \boldsymbol{I})$ denotes the measurement noise. We assume that $\boldsymbol{x}$, $\boldsymbol{z}$, and $\mathbf{H}$ are independently drawn from their respective distributions for each instance of the problem.

To simplify our analysis, we consider the singular value decomposition (SVD) to the measurement operator $\mathbf{H}$ (Kawar et al., 2022; 2023). This decomposition facilitates a transformation that decouples

the measurement process and allows the KL divergence—originally defined in the image domain—to be re-expressed in the measurement domain. We write the SVD of $\mathbf{H}$ as

$$\mathbf{H} = \boldsymbol{U}\boldsymbol{\Sigma}\boldsymbol{V}^{\mathsf{T}}, \tag{6}$$

where $\boldsymbol{U} \in \mathbb{R}^{m \times m}$ and $\boldsymbol{V} \in \mathbb{R}^{n \times n}$ are orthogonal matrices, and $\boldsymbol{\Sigma} \in \mathbb{R}^{m \times n}$ is a matrix of singular values. We define three transformed variables: $\bar{\boldsymbol{x}} = \boldsymbol{V}^{\mathsf{T}}\boldsymbol{x}$, $\bar{\boldsymbol{y}} = \boldsymbol{\Sigma}^{\dagger}\boldsymbol{U}^{\mathsf{T}}\boldsymbol{y}$, and $\bar{\boldsymbol{z}} = \boldsymbol{\Sigma}^{\dagger}\boldsymbol{U}^{\mathsf{T}}\boldsymbol{z}$. Substitution of these variables into the original measurement model in Eq. (5), leads to relationship

$$\bar{\boldsymbol{y}} = \boldsymbol{P}\bar{\boldsymbol{x}} + \bar{\boldsymbol{z}}, \tag{7}$$

where $\boldsymbol{P} = \boldsymbol{\Sigma}^{\dagger}\boldsymbol{\Sigma}$ is a diagonal projection matrix with entries in $\{0, 1\}$, and $\bar{\boldsymbol{z}} \sim \mathcal{N}(0, \sigma_{\boldsymbol{z}}^2 \boldsymbol{\Sigma}^{\dagger}\boldsymbol{\Sigma}^{\dagger\mathsf{T}})$ represents anisotropic uncorrelated Gaussian noise.

In the noiseless setting, we can rewrite Eq. (7) as $\bar{\boldsymbol{y}} = \boldsymbol{P}\bar{\boldsymbol{x}}$. For every noise level $\sigma$ in the noise schedule vector $\boldsymbol{\sigma}$, we consider a noisy version of the SVD observations

$$\bar{\boldsymbol{y}}_\sigma = \boldsymbol{P}\bar{\boldsymbol{x}}_\sigma = \boldsymbol{P}\bar{\boldsymbol{x}} + \bar{\boldsymbol{n}} = \bar{\boldsymbol{y}} + \bar{\boldsymbol{n}}, \qquad \text{where} \qquad \bar{\boldsymbol{n}} = \boldsymbol{P}\boldsymbol{n} \sim \mathcal{N}(0, \sigma^2 \boldsymbol{P}), \tag{8}$$

where $\boldsymbol{n} \sim \mathcal{N}(0, \sigma^2 \boldsymbol{I})$ and $\boldsymbol{P}$ is an orthogonal projection. Note that $\boldsymbol{z}$ refers to measurement noise in inverse problems, while $\boldsymbol{n}$ denotes noise added in the diffusion process.

## 3.2 THEORETICAL RESULTS

We now present our main theoretical result for measuring the distribution shift between the InD prior $p(\boldsymbol{x})$ and OOD prior $q(\boldsymbol{x})$. We require the following assumptions to establish our theoretical results.

**Assumption 1.** The range of the measurement operators $\mathbf{H} \sim p(\mathbf{H})$, used across experiments collectively spans the signal space $\mathbb{R}^n$.

The assumption enforces that, on average, the measurement operators collectively observe every signal direction—formally $\mathbb{E}[\boldsymbol{P}]$ is full-rank on the relevant subspace. This assumption is commonly adopted in self-supervised inverse problems (Kawar et al., 2023; Aggarwal et al., 2022).

**Assumption 2.** Let $\mathbf{H} \sim p(\mathbf{H})$ be a random measurement operator drawn from the family $\mathbf{H}$, and define $\boldsymbol{\Pi} := \mathbf{H}^{\dagger}\mathbf{H}$. For each diffusion noise level $\sigma$, the denoiser residuals $\mathsf{D}(\boldsymbol{V}\bar{\boldsymbol{y}}_\sigma) - \boldsymbol{V}\bar{\boldsymbol{y}}_\sigma$ and $\widehat{\mathsf{D}}(\boldsymbol{V}\bar{\boldsymbol{y}}_\sigma) - \boldsymbol{V}\bar{\boldsymbol{y}}_\sigma$ are statistically independent of $\mathbf{H}$ and hence of $\boldsymbol{\Pi}$, where $\mathsf{D}$ and $\widehat{\mathsf{D}}$ denote InD and OOD models, respectivley.

This assumption is standard in self-supervised settings, where masks or operators are randomized independently of the denoiser so that residual energy factorizes, yielding unbiased measurement-only objectives. It also appears in frameworks such as ENSURE and GSURE (Kawar et al., 2023; Aggarwal et al., 2022), and has been corroborated empirically in prior work.

**Theorem 1.** *Let $\bar{\boldsymbol{y}}_\sigma = \boldsymbol{P}\bar{\boldsymbol{x}} + \bar{\boldsymbol{n}}$ denote the noisy projected measurements at noise level $\sigma$ according to Eq. (8). Then, the KL divergence between the InD density $p(\boldsymbol{x})$ and the OOD density $q(\boldsymbol{x})$ can be expressed as*

$$\begin{aligned}
\mathrm{D_{KL}}(p(\boldsymbol{x}) \,\|\, q(\boldsymbol{x})) = &\int_0^\infty \mathbb{E}\big[\|\boldsymbol{W}(\widehat{\mathsf{D}}_\sigma(\boldsymbol{V}\bar{\boldsymbol{y}}_\sigma) - \boldsymbol{V}\bar{\boldsymbol{y}}_\sigma)\|_2^2\big]\,\sigma^{-3}\,\mathrm{d}\sigma \\
&- \int_0^\infty \mathbb{E}\big[\|\boldsymbol{W}(\mathsf{D}_\sigma(\boldsymbol{V}\bar{\boldsymbol{y}}_\sigma) - \boldsymbol{V}\bar{\boldsymbol{y}}_\sigma)\|_2^2\big]\,\sigma^{-3}\,\mathrm{d}\sigma,
\end{aligned} \tag{9}$$

*where $\mathsf{D}(\boldsymbol{x}_\sigma) = \mathbb{E}_p[\boldsymbol{x}|\boldsymbol{x}_\sigma]$ is the InD model, $\widehat{\mathsf{D}}(\sigma) = \mathbb{E}_q[\boldsymbol{x}|\boldsymbol{x}_\sigma]$ is the OOD model, $\boldsymbol{W}^2 = \mathbb{E}[\mathbf{H}^{\dagger}\mathbf{H}]$ is a scaling matrix, $\boldsymbol{V}$ is the right singular vector from SVD of $\mathbf{H}$, and expectation is taken over $\mathbf{H}$ and $\bar{\boldsymbol{y}} \sim p(\bar{\boldsymbol{y}}|\mathbf{H})$.*

Theorem 1 shows that KL divergence can be computed entirely in the measurement domain, without access to clean images. The integrand depends only on discrepancies between InD and OOD denoisers evaluated at noisy projected measurements $\boldsymbol{V}\bar{\boldsymbol{y}}_\sigma$. This allows us to quantify distribution shift using only the observed measurements, the known forward operator, and pretrained score functions—*without access to clean images*. The proof of Theorem 1 is provided in Appendix B.

When no InD model is available, the KL in Theorem 1 cannot be computed; however, the metric can still be used for model selection. By Theorem 1, for any candidate OOD model in a pool, the KL

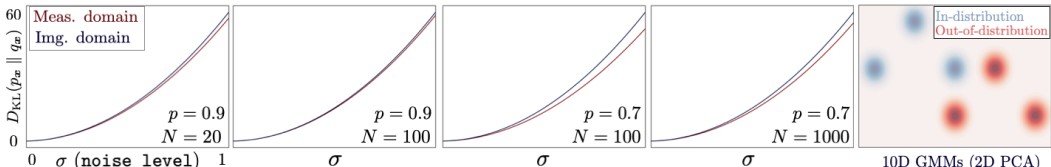

Figure 1: *KL divergence plotted against the noise level $\sigma$ for InD and OOD Gaussian mixture models (GMMs). KL divergence computed in the image domain (blue) and measurement domain (red) under inpainting corruption with probability $p$, using $N$ InD data example. The measurement-domain KL divergence closely tracks its image-domain counterpart, and the approximation improves with increasing $N$ and $p$.*

decomposes into the difference of two "compound MSE" terms aggregated over noise levels: (i) the compound MSE computed with the candidate OOD model, and (ii) the same quantity computed with the (fixed) InD model. The second term is the same for all candidates because it depends only on the true InD distribution. Therefore, minimizing the OOD compound MSE over the pool is equivalent to minimizing the KL up to an additive constant. We exploit this equivalence to pick the OOD prior that is closest to the InD distribution for a given measurement set.

The weighting matrix $W$ is introduced to compensate for the effect of the measurement matrix $\mathbf{H}$, ensuring that all components contribute proportionally—particularly when the likelihood of different $\mathbf{H}$ realizations is imbalanced. The accuracy of the KL approximation is directly tied to the quality of the expectation estimates, which depends on the number of example measurements $N$ used in the computation. As $N$ increases, the empirical estimate of the expectation becomes more reliable, leading to a tighter approximation of the KL divergence. Figure 1 illustrates this relationship using a toy example with Gaussian mixture models (GMMs), where the KL divergence between InD and OOD distributions is plotted as a function of the diffusion noise level $\sigma$. The blue curve represents the KL divergence computed in the image domain, while the red curve shows the corresponding approximation in the measurement domain under inpainting corruption with probability $p$. As shown, the measurement-domain KL closely tracks its image-domain counterpart, validating the effectiveness of our proposed metric under varying levels of measurement corruption.

## 4 EXPERIMENTS

We evaluate our framework on two representative inverse problems: MRI reconstruction and image inpainting. The experiments cover three aspects of our method: model selection, estimation of KL divergence, and adaptation to reduce distribution shifts. To further test robustness, we also study JPEG compression, a setting that violates our theoretical assumptions. Across all tasks, the proposed metric consistently identifies the most suitable model, provides reliable KL divergence estimates without requiring clean images, and supports effective adaptation. Even under JPEG compression, the metric remains practical and reliable, demonstrating robustness beyond the idealized conditions of our theory.

**Inpainting.** For the inpainting experiments, we use FFHQ (Karras et al., 2019) as the InD dataset, with a diffusion model trained to approximate its score function. OOD models are trained separately on AFHQ (Choi et al., 2020), MetFaces(Karras et al., 2020), and Microscopy (CHAMMI) (Chen et al., 2023). Following the protocol of Kawar et al. (2023), images are resized to $64 \times 64$, divided into non-overlapping $4 \times 4$ patches, and each patch is randomly erased with probability $p$. All diffusion models are trained with the framework of Karras et al. (2022). We also use the same model for JPEG compression with quality factors QF $\in \{10, 30, 50, 80\}$.

**MRI.** For MRI experiments, we use the fastMRI dataset (Knoll & et. al, 2020; Zbontar & et. al., 2019). Brain MRI scans are treated as InD data, and a diffusion model is trained on center-cropped $320 \times 320$ slices to approximate their score function. OOD models are trained on knee and prostate MRI slices from the same dataset. To simulate accelerated MRI, we follow established protocols in Kawar et al. (2023); Jalal et al. (2021), applying Cartesian undersampling masks with acceleration factors $R \in \{4, 6, 8\}$, where high-frequency components are sampled randomly.

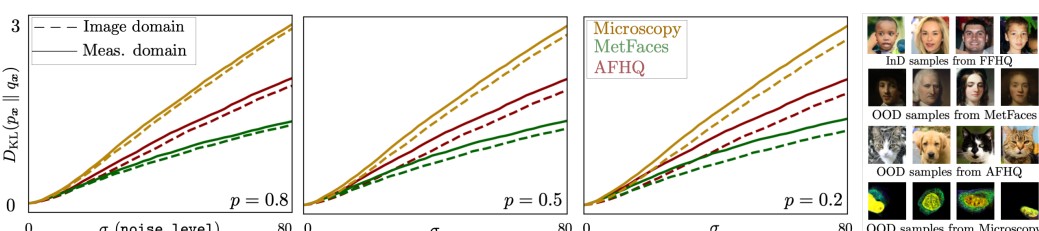

Figure 2: *Comparison of the distribution shift (dashed lines), computed using clean images, and our proposed measurement-domain KL metric (solid lines) between an InD model trained on FFHQ and OOD models trained on MetFaces, AFHQ, and Microscopy. Results are shown under inpainting masks with $p \in \{0.2, 0.5, 0.8\}$. The vertical axis shows $D_{\mathrm{KL}}$, evaluated as the integrand in Eq. (9) and Eq. (4) up to diffusion noise level $\sigma$. Right: Samples from InD and OOD datasets. Note how the proposed metric accurately tracks the KL divergence, even under high-levels of corruption (smaller values of $p$).*

## 4.1 MODEL SELECTION

We first evaluate our metric in the setting of unsupervised model selection, where only corrupted measurements and a pool of pretrained diffusion models are available. For each candidate model, we compute the compound MSE residual $\int_0^\infty \mathbb{E}[\|\boldsymbol{W}(\widehat{\mathsf{D}}_\sigma(\boldsymbol{V}\bar{\boldsymbol{y}}_\sigma) - \boldsymbol{V}\bar{\boldsymbol{y}}_\sigma)\|_2^2]\,\sigma^{-3}\,\mathrm{d}\sigma$, which measures how well the model's score function aligns with the observed measurements. Because the InD contribution is constant across all candidates, minimizing this residual is equivalent to minimizing the KL divergence. Thus, the model with the smallest residual is the one whose prior distribution is closest to the test data. This provides a simple and effective procedure for selecting among pretrained diffusion models directly from corrupted measurements, without requiring clean images.

Table 1 reports the model selection metric under both inpainting masks and JPEG compression. In all settings, the InD (FFHQ) achieves the lowest residual, while OOD models yield higher values. Among the OOD candidates, MetFaces is consistently ranked closest to FFHQ, reflecting its greater visual similarity to the InD data. These results confirm that the compound MSE residual is a reliable model selection metric when only corrupted measurements are available. Notably, the robustness extends to JPEG compression, where despite structured artifacts violating the assumptions of Theorem 1, the metric produces meaningful rankings.

## 4.2 COMPUTING DISTRIBUTION SHIFT

We next evaluate how well the proposed metric estimates KL divergence directly from corrupted measurements. Using Theorem 1, we compute the measurement-domain KL and compare it with reference values obtained in the image domain Eq. (4). When the diffusion models accurately capture the underlying score functions, the image-domain KL provides a ground-truth baseline. This comparison allows us to assess how closely the measurement-domain estimator tracks the true distributional shift.

Figure 2 and Figure 4 compare measurement-domain and image-domain KL divergence for inpainting and MRI tasks. In both cases, the measurement-domain KL closely tracks its image-domain counterpart, confirming that the proposed estimator provides an accurate proxy for distributional shift without requiring clean images. For inpainting, the relative ordering of models is preserved across all mask rates, with MetFaces consistently closest to the InD model (FFHQ), demonstrating the least amount of KL divergence. For MRI, brain scans serve as the InD data, and the estimator reliably distinguishes them from knee and prostate scans under accelerated subsampling, mirroring the image-domain KL. Additional results under JPEG compression are provided in the supplement Figure 10, where despite structured quantization artifacts violating the assumptions of Theorem 1, the estimator continues to produce meaningful distribution shift. This demonstrate the robustness of the proposed metric beyond the idealized conditions of the theory.

Table 1: *Compound MSE for JPEG compression (left) and inpainting settings (right). In both cases, the InD model FFHQ achieves the lowest MSE, while MetFaces consistently ranks closest among OOD models.*

| | InD | OOD models | | | | InD | OOD models | | |
|---|---|---|---|---|---|---|---|---|---|
| QF | **FFHQ** | **MetFaces** | **AFHQ** | **Microscopy** | $p$ | **FFHQ** | **MetFaces** | **AFHQ** | **Microscopy** |
| 10 | 1.792 | 3.215 | 3.614 | 4.452 | 0.9 | 1.677 | 3.561 | 4.009 | 4.770 |
| 30 | 1.807 | 3.232 | 3.629 | 4.470 | 0.8 | 1.805 | 3.651 | 4.076 | 4.841 |
| 50 | 1.814 | 3.238 | 3.634 | 4.476 | 0.7 | 1.929 | 3.713 | 4.154 | 4.922 |
| 80 | 1.829 | 3.252 | 3.649 | 4.491 | 0.5 | 2.067 | 3.730 | 4.186 | 4.968 |

## 4.3 ADAPTATION EFFECT ON DISTRIBUTION SHIFT

Theorem 1 expresses the KL divergence between InD and OOD data in terms of their compound MSE residuals on the measurements. This observation motivates adapting an OOD model using only projected measurements from the target distribution: by reducing the residual error, we directly decrease the KL divergence and thereby mitigate distribution shift. Given a pretrained OOD denoiser $\widehat{\mathsf{D}}_\sigma$, we define the adaptation loss as

$$\mathcal{L}_{\text{adapt}} = \mathbb{E}_{\boldsymbol{y}, \boldsymbol{y}_\sigma, \sigma}\left[\|\boldsymbol{W}(\widehat{\mathsf{D}}(\boldsymbol{V}\bar{\boldsymbol{y}}_\sigma) - \boldsymbol{V}\bar{\boldsymbol{y}})\|_2^2\right]. \tag{10}$$

where the expectation is taken over measurement $\boldsymbol{y}$ from InD, their noisy counterparts $\boldsymbol{y}_\sigma$, and diffusion noise levels $\sigma$. Minimizing this loss encourages the OOD denoiser to match the InD score function on observed projections, thereby reducing distribution shift without requiring clean images.

To validate the adaptation approach, we start from a OOD model trained on AFHQ and adapt it using only projected measurements from the InD dataset FFHQ. Concretely, we select either $64$ or $128$ FFHQ training images, obtain their corrupted measurements, and project them onto the measurement-defined latent basis $\boldsymbol{V}\bar{\boldsymbol{y}}$. Figure 5 plots the resulting KL divergence across diffusion noise levels $\sigma$, comparing the original AFHQ model to two adapted variants, Adapted64 and Adapted128. Even with this limited adaptation data, the KL curves drop noticeably below the unadapted baseline, with Adapted128 yielding the largest reduction. These results confirm that modest, measurement-only adaptation effectively shrinks distribution shift. These adapted models are obtained by minimizing the measurement-domain KL estimator using only projected FFHQ measurements, illustrating that our method serves as a principled adaptation loss rather than only a selection criterion. cImplementation details and additional experiments appear in Appendix D.4 and Appendix D.5.

We then evaluate how this reduction in KL translates to improvements on a downstream inverse problem. Using DPS (Chung et al., 2023a) for image inpainting, we compare four models: the InD model (FFHQ), the unadapted OOD model (AFHQ), and the two adapted AFHQ variants. The adaptation procedure fine-tunes the OOD model to better approximate the InD score function using only projected measurements, without access to clean images. In addition to identifying which pretrained prior is closer to the test distribution, the KL estimator can serve as a training loss to improve an OOD prior. Figure 4 demonstrates this: starting from an AFHQ-trained model, aligning its score function with FFHQ measurements reduces the measurement-domain KL and produces the adapted models with better reconstructions. The figure includes inpainting results on a representative FFHQ test image with mask rate $p = 0.8$ and measurement noise level $\sigma_{\boldsymbol{z}} = 0.01$, reporting both PSNR and LPIPS. As expected, the unadapted OOD model performs poorly on the InD data, while adaptation with $64$ or $128$ projected measurements substantially improves reconstruction quality. Table 2 confirms this trend quantitatively across settings: adapted models consistently outperform the unadapted OOD model, narrowing the gap toward the InD baseline. These findings show that even lightweight, measurement-based adaptation is effective to mitigate distribution shift in practice.

## 4.4 ABLATION STUDIES

We study how varying the inpainting measurement probability, which controls the degree of ill-posedness, influences the accuracy of KL divergence approximation using the proposed metric. Figure 6 illustrates the difference between the KL divergence computed on clean images and our metric obtained from measurements masked with varying inpainting probabilities. As expected, lower measurement corruption (i.e., higher sampling probability) leads to more accurate KL divergence

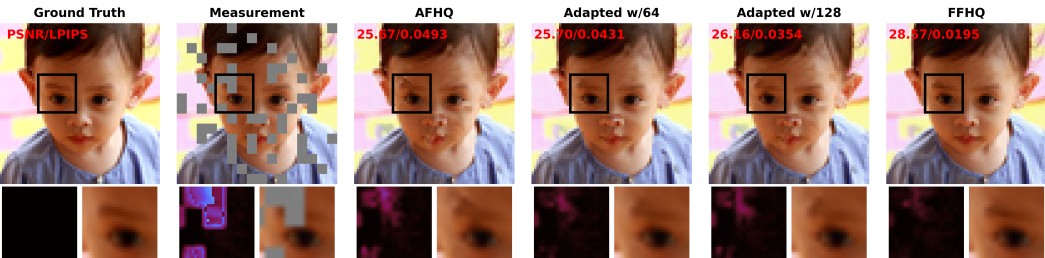

Figure 3: *Visual comparison of inpainting results (DPS (Chung et al., 2023a)) on an FFHQ image with mask rate $p = 0.8$ and measurement noise level $\sigma = 0.01$. The top row shows full reconstructions, while the bottom row displays residual maps (left) and zoomed-in regions (right). Note the performance gap between the InD and OOD models, and the improvement achieved by adapting the OOD models using only corrupted measurements.*

Table 2: *Comparison of InD, OOD, and Adapted models for image reconstruction using DPS (Chung et al., 2023a), for inpainting with different inpainting masks and measurement noise.* **Best** *and* **second best** *are shown.*

| Method | $p = 0.8$ $\sigma_{\boldsymbol{z}} = 0.01$ | | $p = 0.9$ $\sigma_{\boldsymbol{z}} = 0.00$ | |
|---|---|---|---|---|
| | PSNR ↑ | LPIPS↓ | PSNR↑ | LPIPS↓ |
| Microscopy | 21.68 | 0.1466 | 25.14 | 0.0707 |
| MetFaces | 25.49 | 0.0766 | 29.60 | 0.0342 |
| AFHQ | 25.84 | 0.0614 | 30.02 | 0.0246 |
| FFHQ | **28.36** | **0.0322** | **33.24** | **0.0113** |
| Adapt64 (AFHQ) | 26.14 | 0.0530 | 30.23 | 0.0208 |
| Adapt128 (AFHQ) | **26.52** | **0.0465** | **30.37** | **0.0187** |

estimates. However, the proposed metric remains effective in providing an approximation of the image-domain KL divergence, even under high levels of measurement corruption.

Table 3 presents an ablation study analyzing how the KL divergence approximation responds to measurement noise $\sigma_{\boldsymbol{z}}$ and number of measurement examples $N$ from InD dataset. The results show that KL estimates remain stable even under substantial noise, supporting the robustness of the proposed metric and validating Theorem 1. Notably, reliable estimates are obtained with as few as 20 samples, demonstrating the metric's effectiveness in both noisy and noiseless setting, using limited number of measurement examples from InD dataset.

## 5 CONCLUSION

We introduced a principled framework for measuring distribution shift in inverse problems using only corrupted measurements and pretrained diffusion scores. Theoretically, we derived a measurement-domain KL estimator that connects distribution shift to a compound MSE residual aggregated over diffusion noise levels (Theorem 1). Practically, the same quantity yields a simple model-selection metric when only a pool of priors is available, and it guides lightweight adaptation by aligning score functions on observed projections. Experiments on image inpainting and MRI show that the measurement-domain KL tracks image-domain KL closely and that the proposed residual consistently ranks models in accordance with the test distribution; a JPEG stress test further suggests robustness even when

Figure 4: *Comparison of the distribution shift (dashed lines), computed using clean images, and our proposed measurement-domain KL metric (solid lines) between an InD model trained on Brain slices and OOD models trained on Knee and Prostate slices from fastMRI dataset with acceleration rate 4. The vertical axis shows $D_{\mathrm{KL}}$, evaluated as the integrand in Eq. (9) and Eq. (4) up to diffusion noise level $\sigma$. The proposed metric accurately tracks the KL divergence.*

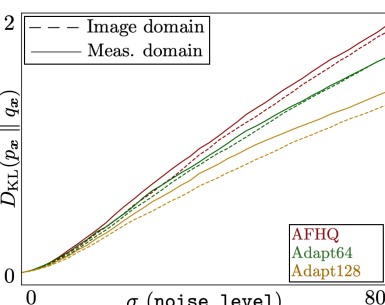

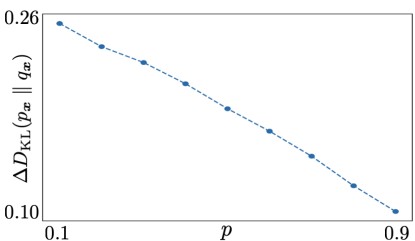

Figure 5: *KL divergence between FFHQ and AFHQ, along with adapted models using 64 and 128 projected measurements. Values are computed in the image domain (dashed) and measurement domain (solid) under inpainting with $p = 0.8$. Adaptation using only projected measurements significantly reduces the distributional gap.*

Figure 6: *Difference between image-domain KL divergence (FFHQ vs. AFHQ) and the measurement-domain approximation across varying inpainting probabilities. Smaller differences indicate better approximation; accuracy improves as corruption decreases, while robustness is maintained under severe corruption.*

Table 3: *KL divergence as a function of data examples $N$ and measurement noise level $\sigma_z$. Note the robustness of the metric to measurement noise. Also note that limited number of corrupted measurement can approximate KL divergence.*

| $\sigma_z$ $N$ | 0.0 | 0.1 | 0.2 | 0.5 | 1.0 | $D_{\mathrm{KL}}$ (Img) |
|---|---|---|---|---|---|---|
| **20** | 2.098 | 2.085 | 2.085 | 2.091 | 2.114 | 1.974 |
| **40** | 2.070 | 2.074 | 2.074 | 2.079 | 2.102 | 1.935 |
| **80** | 2.063 | 2.116 | 2.116 | 2.119 | 2.140 | 1.978 |
| **120** | 2.073 | 2.098 | 2.098 | 2.102 | 2.124 | 1.956 |

theoretical assumptions are violated. Together, these results indicate that distribution shift can be detected, quantified, and mitigated directly from measurements, without access to clean images, and with practical benefits for downstream reconstruction quality.

## LIMITATIONS

Our framework relies on two assumptions. First, the theoretical guarantees require randomized measurement operators whose collective range spans the signal space. While this is standard in self-supervised inverse problems and satisfied by many random sampling schemes, structured operators such as fixed MRI masks may not fully meet this condition; extending the analysis to such settings is an important direction for future work. Second, we assume independence between measurement operators and denoiser residuals. This assumption underpins prior frameworks such as GSURE and ENSURE, but in scenarios where operators are tightly coupled to the training process, it may be violated and introduce bias. Developing techniques that relax this requirement could enhance the robustness of our metric.

## REPRODUCIBILITY STATEMENT

We have released the code, pretrained models, and dataset used for reporting the results.

## LLMS USAGE STATEMENT

We used large language models for editorial assistance. LLMs were not used to design methods, derive theory, run or analyze experiments.

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

## A PROOF OF KL DIVERGENCE METRIC ON IMAGE DOMAIN

The following proof and Eq. (4) results from Theorem 1 of (Song et al., 2021) and it is also briefly discussed in (Kadkhodaie et al., 2024). Let $\nabla_{\boldsymbol{x}_\sigma} \log p(\boldsymbol{x}_\sigma)$ and $\nabla_{\boldsymbol{x}_\sigma} \log q(\boldsymbol{x}_\sigma)$ represent the score of InD $p(\boldsymbol{x})$ and OOD $q(\boldsymbol{x})$, respectively. The distribution shift measured by KL divergence between density functions $p(\boldsymbol{x})$ and $q(\boldsymbol{x})$ can be obtained as

$$D_{\mathrm{KL}}(p(\boldsymbol{x}) \parallel q(\boldsymbol{x})) = \int_0^\infty \mathbb{E}_{\boldsymbol{x} \sim p(\boldsymbol{x}), \boldsymbol{x}_\sigma \sim p(\boldsymbol{x}_\sigma | \boldsymbol{x})} \left[ \|\nabla_{\boldsymbol{x}_\sigma} \log p(\boldsymbol{x}_\sigma) - \nabla_{\boldsymbol{x}_\sigma} \log q(\boldsymbol{x}_\sigma)\|_2^2 \right] \sigma \, \mathrm{d}\sigma. \quad (11)$$

*Proof.* Using the fact that $\boldsymbol{x} = \boldsymbol{x}_{\sigma_0}$, we have

$$\begin{aligned}
&D_{\mathrm{KL}}(p(\boldsymbol{x}) \parallel q(\boldsymbol{x})) \\
&= D_{\mathrm{KL}}(p(\boldsymbol{x}_{\sigma_0}) \parallel q(\boldsymbol{x}_{\sigma_0})) - D_{\mathrm{KL}}(p(\boldsymbol{x}_{\sigma_\infty}) \parallel q(\boldsymbol{x}_{\sigma_\infty})) + D_{\mathrm{KL}}(p(\boldsymbol{x}_{\sigma_\infty}) \parallel q(\boldsymbol{x}_{\sigma_\infty})) \quad (12) \\
&= \int_\infty^0 \frac{\partial D_{\mathrm{KL}}(p(\boldsymbol{x}_\sigma) \parallel q(\boldsymbol{x}_\sigma))}{\partial \sigma} \mathrm{d}\sigma, \quad (13)
\end{aligned}$$

where in the last line, we used the fundamental theorem of calculus and in the last line, we used the fact that $p(\boldsymbol{x}_{\sigma_\infty}) = q(\boldsymbol{x}_{\sigma_\infty}) \approx \mathcal{N}(0, \boldsymbol{I})$.

We calculate the derivative of $D_{\mathrm{KL}}(p(\boldsymbol{x}_\sigma) \parallel q(\boldsymbol{x}_\sigma))$ using chain and quotient rule as:

$$\begin{aligned}
\frac{\partial D_{\mathrm{KL}}(p(\boldsymbol{x}_\sigma) \parallel q(\boldsymbol{x}_\sigma))}{\partial \sigma} &= \frac{\partial}{\partial \sigma} \int_{\mathbb{R}^n} p(\boldsymbol{x}_\sigma) \log \frac{p(\boldsymbol{x}_\sigma)}{q(\boldsymbol{x}_\sigma)} \mathrm{d}\boldsymbol{x}_\sigma \\
&= \int \frac{\partial p(\boldsymbol{x}_\sigma)}{\partial \sigma} \log \frac{p(\boldsymbol{x}_\sigma)}{q(\boldsymbol{x}_\sigma)} \mathrm{d}\boldsymbol{x}_\sigma + \int \frac{\partial p(\boldsymbol{x}_\sigma)}{\partial \sigma} \mathrm{d}\boldsymbol{x}_\sigma - \int \frac{\partial q(\boldsymbol{x}_\sigma)}{\partial \sigma} \frac{p(\boldsymbol{x}_\sigma)}{q(\boldsymbol{x}_\sigma)} \mathrm{d}\boldsymbol{x}_\sigma \\
&= \int \frac{\partial p(\boldsymbol{x}_\sigma)}{\partial \sigma} \log \frac{p(\boldsymbol{x}_\sigma)}{q(\boldsymbol{x}_\sigma)} \mathrm{d}\boldsymbol{x}_\sigma - \int \frac{\partial q(\boldsymbol{x}_\sigma)}{\partial \sigma} \frac{p(\boldsymbol{x}_\sigma)}{q(\boldsymbol{x}_\sigma)} \mathrm{d}\boldsymbol{x}_\sigma, \quad (14)
\end{aligned}$$

where in the last line, we used the fact that $\int p(\boldsymbol{x}_\sigma) \mathrm{d}\boldsymbol{x}_\sigma = 1$.

From Fokker-Planck equation for $n$-dimensional vector $\boldsymbol{x}_\sigma$ for the diffusion coefficient $\sigma$, we have

$$\frac{\partial p(\boldsymbol{x}_\sigma)}{\partial \sigma} = \sigma \nabla_{\boldsymbol{x}_\sigma}^2 p(\boldsymbol{x}_\sigma). \quad (15)$$

Plugging this results in the first term of Eq. (14) yields

$$\begin{aligned}
\int \frac{\partial p(\boldsymbol{x}_\sigma)}{\partial \sigma} \log \frac{p(\boldsymbol{x}_\sigma)}{q(\boldsymbol{x}_\sigma)} \mathrm{d}\boldsymbol{x}_\sigma &= \int \sigma \nabla_{\boldsymbol{x}_\sigma}^2 p(\boldsymbol{x}_\sigma) \log \frac{p(\boldsymbol{x}_\sigma)}{q(\boldsymbol{x}_\sigma)} \mathrm{d}\boldsymbol{x}_\sigma \\
&= \sigma \lim_{\substack{\boldsymbol{a} \to \infty \\ \boldsymbol{b} \to -\infty}} \left[ \nabla_{\boldsymbol{x}_\sigma} p(\boldsymbol{x}_\sigma) \log \frac{p(\boldsymbol{x}_\sigma)}{q(\boldsymbol{x}_\sigma)} \right]_{\boldsymbol{b}}^{\boldsymbol{a}} \\
&\quad - \sigma \int \nabla_{\boldsymbol{x}_\sigma} p(\boldsymbol{x}_\sigma)^\mathsf{T} \left[ \nabla_{\boldsymbol{x}_\sigma} \log p(\boldsymbol{x}_\sigma) - \nabla_{\boldsymbol{x}_\sigma} \log q(\boldsymbol{x}_\sigma) \right] \mathrm{d}\boldsymbol{x}_\sigma \\
&= -\sigma \int \nabla_{\boldsymbol{x}_\sigma} p(\boldsymbol{x}_\sigma)^\mathsf{T} \left[ \nabla_{\boldsymbol{x}_\sigma} \log p(\boldsymbol{x}_\sigma) - \nabla_{\boldsymbol{x}_\sigma} \log q(\boldsymbol{x}_\sigma) \right] \mathrm{d}\boldsymbol{x}_\sigma \\
&= -\sigma \int \nabla_{\boldsymbol{x}_\sigma} \log p(\boldsymbol{x}_\sigma)^\mathsf{T} \left[ \nabla_{\boldsymbol{x}_\sigma} \log p(\boldsymbol{x}_\sigma) - \nabla_{\boldsymbol{x}_\sigma} \log q(\boldsymbol{x}_\sigma) \right] p(\boldsymbol{x}_\sigma) \mathrm{d}\boldsymbol{x}_\sigma, \\
&\quad (16)
\end{aligned}$$

where we used integration by parts and the fact the the first term vanishes when both $p(\boldsymbol{x})$ and $q(\boldsymbol{x})$ and their derivatives decays rapidly at $\pm\infty$. Note that in the last equality, we used the fact that

$\nabla_{\boldsymbol{x}_\sigma} \log p(\boldsymbol{x}_\sigma)p(\boldsymbol{x}_\sigma) = \nabla_{\boldsymbol{x}_\sigma} p(\boldsymbol{x}_\sigma)$. We also have $\frac{\partial q(\boldsymbol{x}_\sigma)}{\partial \sigma} = \sigma\nabla^2_{\boldsymbol{x}_\sigma} q(\boldsymbol{x}_\sigma)$, which yields

$$
\int \frac{\partial q(\boldsymbol{x}_\sigma)}{\partial \sigma}\frac{p(\boldsymbol{x}_\sigma)}{q(\boldsymbol{x}_\sigma)}\mathrm{d}\boldsymbol{x}_\sigma = \int \sigma\nabla^2_{\boldsymbol{x}_\sigma} q(\boldsymbol{x}_\sigma)\frac{p(\boldsymbol{x}_\sigma)}{q(\boldsymbol{x}_\sigma)}\mathrm{d}\boldsymbol{x}_\sigma
$$

$$
= \sigma \lim_{\substack{\boldsymbol{a}\to\infty \\ \boldsymbol{b}\to-\infty}}\left[\nabla_{\boldsymbol{x}_\sigma} q(\boldsymbol{x}_\sigma)\frac{p(\boldsymbol{x}_\sigma)}{q(\boldsymbol{x}_\sigma)}\right]^{\boldsymbol{a}}_{\boldsymbol{b}}
$$

$$
- \sigma\int\nabla_{\boldsymbol{x}_\sigma} q(\boldsymbol{x}_\sigma)^\mathsf{T}\left[\frac{\nabla_{\boldsymbol{x}_\sigma} p(\boldsymbol{x}_\sigma)}{q(\boldsymbol{x}_\sigma)} - \frac{\nabla_{\boldsymbol{x}_\sigma} q(\boldsymbol{x}_\sigma)}{q(\boldsymbol{x}_\sigma)}\frac{p(\boldsymbol{x}_\sigma)}{q(\boldsymbol{x}_\sigma)}\right]\mathrm{d}\boldsymbol{x}_\sigma
$$

$$
= -\sigma\int\nabla_{\boldsymbol{x}_\sigma} q(\boldsymbol{x}_\sigma)^\mathsf{T}\left[\frac{\nabla_{\boldsymbol{x}_\sigma} p(\boldsymbol{x}_\sigma)}{q(\boldsymbol{x}_\sigma)} - \frac{\nabla_{\boldsymbol{x}_\sigma} q(\boldsymbol{x}_\sigma)}{q(\boldsymbol{x}_\sigma)}\frac{p(\boldsymbol{x}_\sigma)}{q(\boldsymbol{x}_\sigma)}\right]\mathrm{d}\boldsymbol{x}_\sigma
$$

$$
= -\sigma\int\nabla_{\boldsymbol{x}_\sigma} q(\boldsymbol{x}_\sigma)^\mathsf{T}\left[\nabla_{\boldsymbol{x}_\sigma} \log p(\boldsymbol{x}_\sigma) - \nabla_{\boldsymbol{x}_\sigma} \log q(\boldsymbol{x}_\sigma)\right]\frac{p(\boldsymbol{x}_\sigma)}{q(\boldsymbol{x}_\sigma)}\mathrm{d}\boldsymbol{x}_\sigma
$$

$$
= -\sigma\int\nabla_{\boldsymbol{x}_\sigma} \log q(\boldsymbol{x}_\sigma)^\mathsf{T}\left[\nabla_{\boldsymbol{x}_\sigma} \log p(\boldsymbol{x}_\sigma) - \nabla_{\boldsymbol{x}_\sigma} \log q(\boldsymbol{x}_\sigma)\right]p(\boldsymbol{x}_\sigma)\mathrm{d}\boldsymbol{x}_\sigma. \tag{17}
$$

Putting Eq. (16) and Eq. (17) in Eq. (14) establishes that

$$
\frac{\partial \mathrm{D}_{\mathrm{KL}}(p(\boldsymbol{x}_\sigma) \parallel q(\boldsymbol{x}_\sigma))}{\partial \sigma} = -\sigma\int p(\boldsymbol{x}_\sigma)\|\nabla_{\boldsymbol{x}_\sigma} \log p(\boldsymbol{x}_\sigma) - \nabla_{\boldsymbol{x}_\sigma} \log q(\boldsymbol{x}_\sigma)\|^2_2\mathrm{d}\boldsymbol{x}_\sigma
$$

$$
= -\sigma\mathbb{E}\left[\|\nabla_{\boldsymbol{x}_\sigma} \log p(\boldsymbol{x}_\sigma) - \nabla_{\boldsymbol{x}_\sigma} \log q(\boldsymbol{x}_\sigma)\|^2_2.\right]
$$

Replacing this equation in Eq. (12) establishes the desired result:

$$
\mathrm{D}_{\mathrm{KL}}(p(\boldsymbol{x}) \parallel q(\boldsymbol{x})) = \int_\infty^0 \frac{\partial \mathrm{D}_{\mathrm{KL}}(p(\boldsymbol{x}_\sigma) \parallel q(\boldsymbol{x}_\sigma))}{\partial \sigma}\mathrm{d}\sigma
$$

$$
= \int_0^\infty \mathbb{E}\left[\|\nabla_{\boldsymbol{x}_\sigma} \log p(\boldsymbol{x}_\sigma) - \nabla_{\boldsymbol{x}_\sigma} \log q(\boldsymbol{x}_\sigma)\|^2_2\right]\sigma\mathrm{d}\sigma. \tag{18}
$$

$\square$

## B    PROOF OF THEOREM 1

**Theorem 1.** *Let $\bar{\boldsymbol{y}}_\sigma = \boldsymbol{P}\bar{\boldsymbol{x}} + \bar{\boldsymbol{n}}$ denote the noisy projected measurements at noise level $\sigma$ according to Eq. (8). Then, the KL divergence between the InD density $p(\boldsymbol{x})$ and the OOD density $q(\boldsymbol{x})$ can be expressed as*

$$
\mathrm{D}_{\mathrm{KL}}(p(\boldsymbol{x}) \parallel q(\boldsymbol{x})) = \int_0^\infty \mathbb{E}\left[\|\boldsymbol{W}(\widehat{\mathsf{D}}_\sigma(\boldsymbol{V}\bar{\boldsymbol{y}}_\sigma) - \boldsymbol{V}\bar{\boldsymbol{y}}_\sigma)\|^2_2\,\sigma^{-3}\,\mathrm{d}\sigma\right.
$$

$$
- \int_0^\infty \mathbb{E}\left[\|\boldsymbol{W}(\mathsf{D}_\sigma(\boldsymbol{V}\bar{\boldsymbol{y}}_\sigma) - \boldsymbol{V}\bar{\boldsymbol{y}}_\sigma)\|^2_2\,\sigma^{-3}\,\mathrm{d}\sigma, \tag{19}
$$

*where $\mathsf{D}(\boldsymbol{x}_\sigma) = \mathbb{E}_p[\boldsymbol{x}|\boldsymbol{x}_\sigma]$ is the InD model, $\widehat{\mathsf{D}}(\sigma) = \mathbb{E}_q[\boldsymbol{x}|\boldsymbol{x}_\sigma]$ is the OOD model, $\boldsymbol{W}^2 = \mathbb{E}[\mathbf{H}^\dagger\mathbf{H}]$ is a scaling matrix, $\boldsymbol{V}$ is the right singular vector from SVD of $\mathbf{H}$, and expectation is taken over $\mathbf{H}$ and $\bar{\boldsymbol{y}} \sim p(\bar{\boldsymbol{y}}|\mathbf{H})$.*

*Proof.* We start from

$$
\mathrm{D}_{\mathrm{KL}}(p(\boldsymbol{x}) \parallel q(\boldsymbol{x})) = \int_0^\infty \mathbb{E}_{\boldsymbol{x},\boldsymbol{x}_\sigma}\left[\|\nabla_{\boldsymbol{x}_\sigma} \log p(\boldsymbol{x}_\sigma) - \nabla_{\boldsymbol{x}_\sigma} \log q(\boldsymbol{x}_\sigma)\|^2_2\right]\sigma\,\mathrm{d}\sigma. \tag{20}
$$

Tweedie's formula states that

$$
\mathsf{D}_p(\boldsymbol{x}_\sigma) = \mathbb{E}_p[\boldsymbol{x}|\boldsymbol{x}_\sigma] = \boldsymbol{x}_\sigma + \sigma^2\nabla\log p(\boldsymbol{x}_\sigma),
$$

By replacing the score in Eq. (20), we have

$$
\mathbb{E}_{\boldsymbol{x},\boldsymbol{x}_\sigma}\left[\|\nabla_{\boldsymbol{x}_\sigma} \log p(\boldsymbol{x}_\sigma) - \nabla_{\boldsymbol{x}_\sigma} \log q(\boldsymbol{x}_\sigma)\|^2_2\right]
$$

$$
= \frac{1}{\sigma^4}\mathbb{E}_{\boldsymbol{x},\boldsymbol{x}_\sigma}\left[\|(\mathbb{E}_p[\boldsymbol{x}|\boldsymbol{x}_\sigma] - \boldsymbol{x}_\sigma) - (\mathbb{E}_q[\boldsymbol{x}|\boldsymbol{x}_\sigma] - \boldsymbol{x}_\sigma)\|^2_2\right]
$$

$$
= \frac{1}{\sigma^4}\left(\mathbb{E}_{\boldsymbol{x},\boldsymbol{x}_\sigma}\left[\|\mathbb{E}_q[\boldsymbol{x}|\boldsymbol{x}_\sigma] - \boldsymbol{x}_\sigma\|^2_2\right] - \mathbb{E}_{\boldsymbol{x},\boldsymbol{x}_\sigma}\left[\|\mathbb{E}_p[\boldsymbol{x}|\boldsymbol{x}_\sigma] - \boldsymbol{x}_\sigma\|^2_2\right]\right), \tag{21}
$$

where in the last line, we used

$$\mathbb{E}_{\boldsymbol{x}, \boldsymbol{x}_\sigma}\left[\|\boldsymbol{x} - \mathbb{E}_q[\boldsymbol{x}|\boldsymbol{x}_\sigma]\|_2^2\right] = \mathbb{E}_{\boldsymbol{x}, \boldsymbol{x}_\sigma}\left[\|\boldsymbol{x} - \mathbb{E}_p[\boldsymbol{x}|\boldsymbol{x}_\sigma] + \mathbb{E}_p[\boldsymbol{x}|\boldsymbol{x}_\sigma] - \mathbb{E}_q[\boldsymbol{x}|\boldsymbol{x}_\sigma]\|_2^2\right]$$

$$= \mathbb{E}_{\boldsymbol{x}, \boldsymbol{x}_\sigma}\left[\|\boldsymbol{x} - \mathbb{E}_p[\boldsymbol{x}|\boldsymbol{x}_\sigma]\|_2^2\right] + \mathbb{E}_{\boldsymbol{x}, \boldsymbol{x}_\sigma}\left[\|\mathbb{E}_p[\boldsymbol{x}|\boldsymbol{x}_\sigma] - \mathbb{E}_q[\boldsymbol{x}|\boldsymbol{x}_\sigma]\|_2^2\right]$$

$$+ 2\mathbb{E}_{\boldsymbol{x}, \boldsymbol{x}_\sigma}\left[\left(\boldsymbol{x} - \mathbb{E}_p[\boldsymbol{x}|\boldsymbol{x}_\sigma]\right)^{\mathsf{T}}\left(\mathbb{E}_p[\boldsymbol{x}|\boldsymbol{x}_\sigma] - \mathbb{E}_q[\boldsymbol{x}|\boldsymbol{x}_\sigma]\right)\right]$$

$$= \mathbb{E}_{\boldsymbol{x}, \boldsymbol{x}_\sigma}\left[\|\boldsymbol{x} - \mathbb{E}_p[\boldsymbol{x}|\boldsymbol{x}_\sigma]\|_2^2\right] + \mathbb{E}_{\boldsymbol{x}, \boldsymbol{x}_\sigma}\left[\|\mathbb{E}_p[\boldsymbol{x}|\boldsymbol{x}_\sigma] - \mathbb{E}_q[\boldsymbol{x}|\boldsymbol{x}_\sigma]\|_2^2\right],$$

In the last equality, we used the fact that

$$\mathbb{E}_{\boldsymbol{x}, \boldsymbol{x}_\sigma}\left[\left(\boldsymbol{x} - \mathbb{E}_p[\boldsymbol{x}|\boldsymbol{x}_\sigma]\right)\right] = 0, \quad \text{where} \quad \boldsymbol{x} \sim p(\boldsymbol{x}) \text{ and } \boldsymbol{x}_\sigma \sim p(\boldsymbol{x}_\sigma).$$

For $\boldsymbol{x}_\sigma = \boldsymbol{x} + \boldsymbol{n}$, where $\boldsymbol{n} \sim \mathcal{N}(0, \sigma^2 \boldsymbol{I})$ is the diffusion process noise. Noting SVD for $\mathbf{H} = \boldsymbol{U}\boldsymbol{\Sigma}\boldsymbol{V}^T$, we define the transformed (right singular vector) coordinates as $\bar{\boldsymbol{x}}_\sigma = \boldsymbol{V}^{\mathsf{T}}\boldsymbol{x} + \boldsymbol{V}^{\mathsf{T}}\boldsymbol{n} = \bar{\boldsymbol{x}} + \boldsymbol{V}^{\mathsf{T}}\boldsymbol{n}$. Since $\boldsymbol{V}^{\mathsf{T}}$ is an orthogonal matrix, the noise remains Gaussian with the same covariance, i.e., $\boldsymbol{V}^{\mathsf{T}}\boldsymbol{n} \sim \mathcal{N}(0, \sigma^2 \boldsymbol{I})$.

Following similar approach to GSURE (Kawar et al., 2023) and (Aggarwal et al., 2022) we define the residual $\boldsymbol{r} \coloneqq \mathsf{D}(\boldsymbol{V}\bar{\boldsymbol{y}}_\sigma) - \boldsymbol{V}\bar{\boldsymbol{y}}_\sigma$. For a given $\boldsymbol{P}$ and $\boldsymbol{V}$, we have

$$\boldsymbol{r} = \mathbb{E}_p[\boldsymbol{V}\bar{\boldsymbol{y}}|\boldsymbol{V}\bar{\boldsymbol{y}}_\sigma, \boldsymbol{V}, \boldsymbol{P}] - \boldsymbol{V}\bar{\boldsymbol{y}}_\sigma$$

$$= \mathbb{E}_p[\boldsymbol{V}\boldsymbol{P}\bar{\boldsymbol{x}}|\boldsymbol{V}\boldsymbol{P}\bar{\boldsymbol{x}}_\sigma, \boldsymbol{V}, \boldsymbol{P}] - \boldsymbol{V}\boldsymbol{P}\bar{\boldsymbol{x}}_\sigma$$

$$= \mathbb{E}_p[\boldsymbol{V}\boldsymbol{P}\boldsymbol{V}^{\mathsf{T}}\boldsymbol{V}\bar{\boldsymbol{x}}|\bar{\boldsymbol{x}}_\sigma, \boldsymbol{V}, \boldsymbol{P}] - \boldsymbol{V}\boldsymbol{P}\boldsymbol{V}^{\mathsf{T}}\boldsymbol{V}\bar{\boldsymbol{x}}_\sigma$$

$$= \mathbb{E}_p[\mathbf{H}^\dagger\mathbf{H}\boldsymbol{V}\bar{\boldsymbol{x}}|\bar{\boldsymbol{x}}_\sigma, \boldsymbol{V}, \boldsymbol{P}] - \mathbf{H}^\dagger\mathbf{H}\boldsymbol{V}\bar{\boldsymbol{x}}_\sigma$$

$$= \mathbb{E}_p[\mathbf{H}^\dagger\mathbf{H}\boldsymbol{x}|\boldsymbol{x}_\sigma, \boldsymbol{V}, \boldsymbol{P}] - \mathbf{H}^\dagger\mathbf{H}\boldsymbol{x}_\sigma \tag{22}$$

We define $\boldsymbol{\Pi} \coloneqq \mathbf{H}^\dagger\mathbf{H}$. By applying this results and using tower rule, we have

$$\mathbb{E}[\|\boldsymbol{W}\boldsymbol{r}\|_2^2] \stackrel{1}{=} \mathbb{E}\left[\text{Tr}\left(\boldsymbol{W}\boldsymbol{r}\boldsymbol{r}^{\mathsf{T}}\boldsymbol{W}\right)\right]$$

$$\stackrel{2}{=} \mathbb{E}\left[\text{Tr}\left(\boldsymbol{W}^2\boldsymbol{r}\boldsymbol{r}^{\mathsf{T}}\right)\right]$$

$$\stackrel{3}{=} \mathbb{E}\left[\text{Tr}\left(\boldsymbol{W}^2\boldsymbol{\Pi}(\mathbb{E}_p[\boldsymbol{x}|\boldsymbol{x}_\sigma] - \boldsymbol{x}_\sigma)(\mathbb{E}_p[\boldsymbol{x}|\boldsymbol{x}_\sigma] - \boldsymbol{x}_\sigma)^{\mathsf{T}}\boldsymbol{\Pi}\right)\right]$$

$$\stackrel{4}{=} \text{Tr}\left(\mathbb{E}\left[\boldsymbol{W}^2\boldsymbol{\Pi}(\mathbb{E}_p[\boldsymbol{x}|\boldsymbol{x}_\sigma] - \boldsymbol{x}_\sigma)(\mathbb{E}_p[\boldsymbol{x}|\boldsymbol{x}_\sigma] - \boldsymbol{x}_\sigma)^{\mathsf{T}}\right]\right)$$

$$\stackrel{5}{=} \text{Tr}\left(\boldsymbol{W}^2\mathbb{E}[\boldsymbol{\Pi}]\mathbb{E}\left[(\mathbb{E}_p[\boldsymbol{x}|\boldsymbol{x}_\sigma] - \boldsymbol{x}_\sigma)(\mathbb{E}_p[\boldsymbol{x}|\boldsymbol{x}_\sigma] - \boldsymbol{x}_\sigma)^{\mathsf{T}}\right]\right)$$

$$\stackrel{6}{=} \text{Tr}\left(\mathbb{E}\left[(\mathbb{E}_p[\boldsymbol{x}|\boldsymbol{x}_\sigma] - \boldsymbol{x}_\sigma)(\mathbb{E}_p[\boldsymbol{x}|\boldsymbol{x}_\sigma] - \boldsymbol{x}_\sigma)^{\mathsf{T}}\right]\right)$$

$$\stackrel{7}{=} \mathbb{E}\left[\text{Tr}\left((\mathbb{E}_p[\boldsymbol{x}|\boldsymbol{x}_\sigma] - \boldsymbol{x}_\sigma)(\mathbb{E}_p[\boldsymbol{x}|\boldsymbol{x}_\sigma] - \boldsymbol{x}_\sigma)^{\mathsf{T}}\right)\right]$$

$$\stackrel{8}{=} \mathbb{E}\left[\|\mathbb{E}_p[\boldsymbol{x}|\boldsymbol{x}_\sigma] - \boldsymbol{x}_\sigma\|_2^2\right] \tag{23}$$

In equality 1, we use the identity $\|\boldsymbol{m}\|_2^2 = \text{Trace}(\boldsymbol{m}\boldsymbol{m}^{\mathsf{T}})$ for any vector $\boldsymbol{m}$.

In equality 2, we apply the cyclic invariance of the trace operator: $\text{Trace}(\boldsymbol{X}\boldsymbol{Y}\boldsymbol{Z}) = \text{Trace}(\boldsymbol{Z}\boldsymbol{X}\boldsymbol{Y})$.

In equality 3 and 5, we used the independence of denoiser residual from $\mathbf{H}$ in assumption 2 and the result from Eq. (22).

In equality 4, we used the fact that $\boldsymbol{\Pi}\boldsymbol{\Pi} = \boldsymbol{\Pi}$. In equality 6, we used the fact that $\boldsymbol{W}^2 = \mathbb{E}[\mathbf{H}^\dagger\mathbf{H}]$.

Note that the result of Eq. (23) states that

$$\mathbb{E}[\|\boldsymbol{W}(\mathsf{D}(\boldsymbol{V}\bar{\boldsymbol{y}}_\sigma) - \boldsymbol{V}\bar{\boldsymbol{y}}_\sigma)\|_2^2] = \mathbb{E}\left[\|\mathbb{E}_p[\boldsymbol{x}|\boldsymbol{x}_\sigma] - \boldsymbol{x}_\sigma\|_2^2\right]. \tag{24}$$

Similarly, we have

$$\mathbb{E}[\|\boldsymbol{W}(\widehat{\mathsf{D}}(\boldsymbol{V}\bar{\boldsymbol{y}}_\sigma) - \boldsymbol{V}\bar{\boldsymbol{y}}_\sigma)\|_2^2] = \mathbb{E}\left[\|\mathbb{E}_q[\boldsymbol{x}|\boldsymbol{x}_\sigma] - \boldsymbol{x}_\sigma\|_2^2\right]. \tag{25}$$

Combining the last two equations with the result of Eq. (20) and Eq. (21), we have

$$
\begin{aligned}
D_{\mathrm{KL}}(p(\boldsymbol{x}) \parallel q(\boldsymbol{x})) &= \int_0^\infty \mathbb{E}_{\boldsymbol{x},\boldsymbol{x}_\sigma}\left[\|\nabla_{\boldsymbol{x}_\sigma}\log p(\boldsymbol{x}_\sigma) - \nabla_{\boldsymbol{x}_\sigma}\log q(\boldsymbol{x}_\sigma)\|_2^2\right]\,\sigma\,\mathrm{d}\sigma \\
&= \int_0^\infty \left(\mathbb{E}_{\boldsymbol{x},\boldsymbol{x}_\sigma}\left[\|\mathbb{E}_q[\boldsymbol{x}|\boldsymbol{x}_\sigma] - \boldsymbol{x}_\sigma\|_2^2\right] - \mathbb{E}_{\boldsymbol{x},\boldsymbol{x}_\sigma}\left[\|\mathbb{E}_p[\boldsymbol{x}|\boldsymbol{x}_\sigma] - \boldsymbol{x}_\sigma\|_2^2\right]\right)\sigma^{-3}\,\mathrm{d}\sigma \\
&= \int_0^\infty \left(\mathbb{E}\left[\|\boldsymbol{W}(\widehat{\mathsf{D}}(\boldsymbol{V}\bar{\boldsymbol{y}}_\sigma) - \boldsymbol{V}\bar{\boldsymbol{y}}_\sigma)\|_2^2\right] - \mathbb{E}\left[\|\boldsymbol{W}(\mathsf{D}(\boldsymbol{V}\bar{\boldsymbol{y}}_\sigma) - \boldsymbol{V}\bar{\boldsymbol{y}}_\sigma)\|_2^2\right]\right)\sigma^{-3}\,\mathrm{d}\sigma \\
&= \int_0^\infty \mathbb{E}\left[\|\boldsymbol{W}(\widehat{\mathsf{D}}(\boldsymbol{V}\bar{\boldsymbol{y}}_\sigma) - \boldsymbol{V}\bar{\boldsymbol{y}}_\sigma)\|_2^2\,\sigma^{-3}\,\mathrm{d}\sigma\right. \\
&\quad - \int_0^\infty \mathbb{E}\left[\|\boldsymbol{W}(\mathsf{D}(\boldsymbol{V}\bar{\boldsymbol{y}}_\sigma) - \boldsymbol{V}\bar{\boldsymbol{y}}_\sigma)\|_2^2\,\sigma^{-3}\,\mathrm{d}\sigma,\right.
\end{aligned}
\tag{26}
$$

which is the desired result. In the setting with noise measurement, we have a known measurement noise level $\sigma_{\boldsymbol{z}}$. Consider a noisy measurement $\boldsymbol{y}$ acquired using an imaging system according to $\boldsymbol{y} = \mathbf{H}\boldsymbol{x} + \boldsymbol{z}$, where $\boldsymbol{x} \sim p(\boldsymbol{x})$ and $\boldsymbol{z} \sim \mathcal{N}(0, \sigma_{\boldsymbol{z}}^2 \boldsymbol{I})$. Using SVD of $\mathbf{H}$, we have $\bar{\boldsymbol{y}} = \boldsymbol{P}\bar{\boldsymbol{x}} + \bar{\boldsymbol{z}}$, where $\bar{\boldsymbol{z}} = \boldsymbol{\Sigma}^\dagger \boldsymbol{U}^T \boldsymbol{z}$ as in Eq. (7). For every noise level $\sigma$ in noise schedule vector $\boldsymbol{\sigma}$, we create noisy version of SVD observations as

$$
\bar{\boldsymbol{y}}_\sigma = \boldsymbol{P}\bar{\boldsymbol{x}}_\sigma = \boldsymbol{P}\bar{\boldsymbol{x}} + \bar{\boldsymbol{n}} + \bar{\boldsymbol{z}} = \bar{\boldsymbol{y}} + \bar{\boldsymbol{n}}, \qquad \text{where} \qquad \bar{\boldsymbol{n}} = \boldsymbol{P}\boldsymbol{n} \sim \mathcal{N}(0, \sigma^2 \boldsymbol{P}).
\tag{27}
$$

We have the same result for this case as well, since we have the similar relations $\bar{\boldsymbol{y}}_\sigma = \boldsymbol{P}\bar{\boldsymbol{x}}_\sigma$ and $\bar{\boldsymbol{y}} = \boldsymbol{P}\bar{\boldsymbol{x}}$. $\qquad\square$

## C RELATED WORKS

Distribution shift between training and test data distributions is a fundamental challenge in machine learning, with direct implications for model reliability and robustness (Taori et al., 2020; Malinin & et. al, 2021; Zhang et al., 2023; Kulinski & Inouye, 2023). Accurately measuring distribution shift is essential for understanding when models will generalize poorly, and OOD detection techniques often aim to signal such shifts by evaluating feature-based, likelihood-based, or reconstruction-based OOD metrics (Cui & Wang, 2022; Fang et al., 2022; Fort et al., 2021).

Recent works have also explored diffusion-based OOD detection, including perturbation-based score tests, diffusion inpainting detectors, and anomaly-scoring methods using unconditional diffusion models (Heng et al., 2024; Le Bellier & Audebert, 2024; Graham et al., 2023a; Gao et al., 2023; Liu et al., 2023; Livernoche et al., 2024). However, these methods operate directly on clean images and typically formulate OOD detection as a binary image-level classification task, limiting their applicability in settings where only corrupted measurements are available.

To mitigate the impact of distribution shift, adaptation strategies have been developed that modify models post-training to better align with the test distribution (Farahani et al., 2021). In the context of diffusion models, such strategies typically focus on adjusting the generative process, modifying score functions, or fine-tuning to improve robustness against domain shifts (Kang et al., 2019; Ganin & Lempitsky, 2015; You et al., 2019; Sun et al., 2016; Ben-David et al., 2006). While these methods can reduce performance degradation, they often assume access to clean adaptation samples or reconstruction proxies.

In imaging inverse problems, the challenges of distribution shift, OOD detection, and adaptation are amplified by the absence of clean images at test time (Gilton et al., 2021; Yismaw et al., 2024; Shoushtari et al., 2024; Chung & Ye, 2024; Chung et al., 2023b). Conventional approaches to quantifying shift and adapting models are not directly applicable, as only corrupted measurements are available. This motivates the need for measurement-domain metrics and adaptation techniques that operate without requiring ground-truth reconstructions—precisely the setting we address in this work.

# D  IMPLEMENTATION DETAILS

## D.1  INPAINTING

**Dataset.** We use the Flickr-Faces-HQ (FFHQ) dataset (Karras et al., 2019) as our InD data. For OOD data, we include images from the AFHQ (Choi et al., 2020), MetFaces (Karras et al., 2020), and Microscopy (CHAMMI) (Chen et al., 2023) datasets. All images were resized to $64 \times 64$ for training and evaluation.

Test samples are randomly drawn from the FFHQ test set (the last $10,000$ images). For KL divergence experiments (Figure 2), we select 20 images (included in the supplementary materials) and process them using the inpainting measurement model. The same test set is also used for image reconstruction with the DPS algorithm.

For adaptation experiments, we sample random images from the FFHQ training set. When required by the diffusion models, data is normalized to the $[-1, 1]$ range.

**Model checkpoints.** InD diffusion model for FFHQ and OOD AFHQ were taken from (Karras et al., 2022) (DDPM++ using EDM preconditioning). Similar training strategy was used for microscopy and MetFaces diffusion models one NVIDIA A100 GPUs. All experiments regarding KL divergence were obtained using one NVIDIA RTX A6000 GPU.

**Measurement model.** We followed the inpainting corruption setup from (Kawar et al., 2023), where the degradation operator $\mathbf{H}$ randomly masks non-overlapping $4 \times 4$ patches across each image with probability p, independently per sample. Each $\mathbf{H}$ is a sample-specific binary diagonal matrix that acts element-wise. As a diagonal matrix, $\mathbf{H}$ is symmetric, idempotent, and admits the singular value decomposition $\mathbf{H} = \boldsymbol{I}\boldsymbol{\Sigma}\boldsymbol{I}^\top$, where $\boldsymbol{\Sigma} = \mathbf{H}$ has entries in {0,1}. This implies that the projection matrix $\boldsymbol{P} = \mathbf{H}^\top\mathbf{H}$, and all measurement operators share the same right-singular vectors $\boldsymbol{V} = \boldsymbol{I}$. The stochastic nature of the masking ensures that all pixels are eventually observed across different $\mathbf{H}$, and the union of their row spaces spans $\mathbb{R}^n$, satisfying Assumption 1.

The code, data, and models can be found here [1].

## D.2  FASTMRI

**Datasets.** We use brain MRI images from the fastMRI dataset (Knoll & et. al, 2020; Zbontar & et. al., 2019) as the InD data. All images are center-cropped to a resolution of $320 \times 320$ for training. The training set consists of 48,406 slices, where only slices with index greater than 5 are included. For OOD data, we extract 29,877 slices from single-coil knee MRI scans and 7,673 slices from prostate MRI scans. For evaluation, 20 images from the brain MRI validation set are used as the test set.

**Model checkpoints.** Diffusion models for all three datasets were trained using (Karras et al., 2022) (DDPM++ using EDM preconditioning) using one NVIDIA A100 GPUs. All experiments regarding KL divergence were obtained using one NVIDIA RTX A6000 GPU.

**Measurement model.** We followed the MRI measurement setup from (Jalal et al., 2021; Kawar et al., 2023) to create the corrupted data. The measurement operator $\mathbf{H}$ performs partial Fourier sampling along the frequency (readout) axis, with an acceleration factor $R$. Specifically, $\mathbf{H}$ retains the lowest $120/R$ frequency components and randomly selects an additional $200/R$ frequencies from the remaining spectrum, yielding a total of $320/R$ retained lines out of 320. The operator can be expressed as $\mathbf{H} = \boldsymbol{I}\boldsymbol{\Sigma}\boldsymbol{F}$, where $\boldsymbol{F}$ denotes the discrete Fourier transform and $\boldsymbol{\Sigma}$ is a diagonal binary matrix encoding the sampling pattern. This representation serves as a valid SVD of $\mathbf{H}$ and can be efficiently implemented via FFT. The combination of fixed low-frequency sampling with randomized high-frequency selection ensures that the union of observed frequency components across samples covers the full signal space (satisfying Assumption 1).

---

[1] https://drive.google.com/drive/folders/1VmCcM33gaSZUNSOorKL1FkilI4jOutAf?usp=share_link

## D.3  KL DIVERGENCE EXPERIMENTS ON GMMS

To visualize and validate KL divergence estimation between distributions under varying noise levels and partial observations, we designed a synthetic setup using Gaussian mixture models (GMMs) in a 10-dimensional space. Both the InD and OOD were defined as GMMs with $K = 3$ components, each having equal weights and isotropic Gaussian covariances. The component means of the InD distribution were arranged to form a structured triangular configuration in the first two principal dimensions of $\mathbb{R}^{10}$: component means were placed along the x-axis with offsets of $5$ units, while alternating vertically in the y-direction to create separation. Specifically, the InD means were defined as $[0, 0, 0, \ldots], [5, 5, 0, \ldots], [10, 0, 0, \ldots]$, with the remaining eight dimensions set to zero. All InD components shared identical covariance matrices, set to the $10 \times 10$ identity matrix, yielding isotropic spreads in all directions.

To construct the OOD distribution, each InD component mean was shifted in the first two dimensions: 10 units along the x-axis and $-5$ units along the y-axis. This resulted in OOD component centers that were clearly displaced from their InD counterparts: $[10, -5, 0, \ldots], [15, 0, 0, \ldots], [20, -5, 0, \ldots]$. Covariance matrices for the OOD components were again isotropic and identical to the InD case. This setup ensures that the only difference between InD and OOD distributions lies in their location, allowing for a clean assessment of distributional shift without confounding factors such as varying shape or spread.

We approximated the KL divergence metrics both in data and measurement domain using the corresponding formulas using a Riemann sum over $\sigma \in [0.01, 1.0]$. In a measurement-corrupted scenario, we applied random masking to the data with a given probability, zeroing out entries to simulate missing observations (e.g., similar to inpainting). We then computed the same score-based KL on the masked data and compared it to the full-data KL. Visualizations in Figure 1 include PCA projections of the InD and OOD samples in $2D$, showing clear spatial separation in the first two dimensions. Our results show that the KL divergence computed from partially observed (masked) data closely tracks the divergence computed from clean data for various inpainting probablity $p$ and number of samples used for KL metric computing $N$.

## D.4  ADAPTATION

The training follows the training for diffusion models from (Karras et al., 2022) (DDPM++ using EDM preconditioning), without changing the parameters (only batch number was adjusted based on the number of corrupted measurements used). For each batch, same inpainting/MRI mask was used. Adaptation was done using one NVIDIA RTX A6000 GPU. Data-prepartion for the adaptation follows the same procedure for calculating the KL divergence, noted in sections D.1 and D.2. Adaptation is terminated when (i) the training loss fails to improve by $\geq 0.5\%$ over the last 10 kimg, or (ii) $B$ kimg have been processed ($B = 500$ for 64 measurements, $B = 1000$ for 128).

## D.5  ADDITIONAL EXPERIMENTS

Figure 7 extends our evaluation to the MRI measurements using different subsampling masks, comparing the KL divergence—computed from clean brain MRI slices—with our measurement-domain KL metric, using only undersampled k-space measurements. The InD model is trained on brain MRI data, while the OOD models are trained on knee and prostate scans from the fastMRI dataset. Results are shown for acceleration rates $R \in \{4, 6, 8\}$, with the vertical axis representing the truncated KL divergence integrated up to diffusion noise level $\sigma$. As shown, the proposed metric closely follows the KL divergence across all settings, demonstrating its robustness even under aggressive subsampling. Example slices from each dataset are shown on the right.

Figure 8 demonstrates the effect of model adaptation on reducing distribution shift in the MRI setting. We plot the KL divergence between Brain and Prostate MRI slices, both before and after adapting the OOD model using only 64 projected (corrupted) measurements. Results are shown for an acceleration rate of R = 4, with KL evaluated in both the image domain (dashed) and measurement domain (solid). As shown, adaptation using only projected measurements substantially reduces the KL divergence, confirming the effectiveness of our adaptation strategy in bridging the distributional gap without requiring clean images. Table 4 reports the reconstruction results using the adapted, OOD, and InD

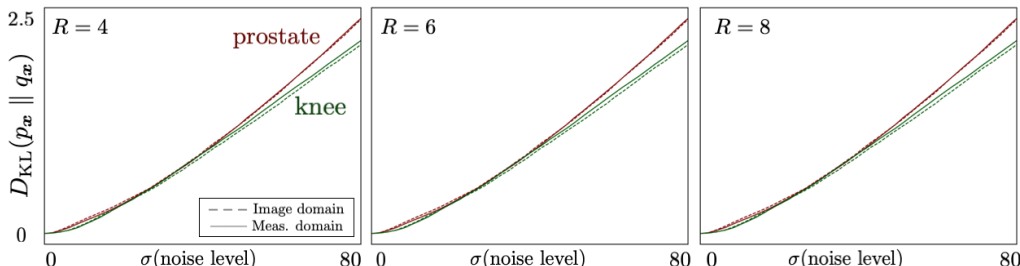

Figure 7: *Comparison of the distribution shift (dashed lines), computed using clean images, and our proposed measurement-domain KL metric (solid lines) between an InD model trained on Brain and OOD models trained on Knee and Prostate MRI slices from fastMRI dataset. Results are shown under MRI acceleration rates $R \in \{4, 6, 8\}$. The vertical axis shows $D_{\mathrm{KL}}$, evaluated as the integrand in Eq. (9 and Eq. (4 up to diffusion noise level $\sigma$. The proposed metric accurately tracks the KL divergence, even under high-levels of corruption. Right: Samples from InD and OOD datasets.*

models. Figure 9 illustrates the visual comparison of reconstruction with DPS using InD, OOD, and Adapted Model.

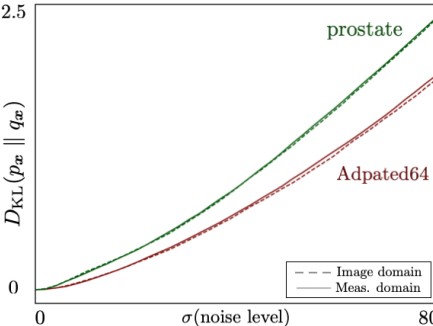

Figure 8: $D_{\mathrm{KL}}$ *between Brain MRI and Prostate MRI, as well as adapted models using 64 projected measurements, measured in the image domain (dashed) and the measurement domain (solid) for subsampled MRI with acceleration rate $R = 4$. Notably, adapting the network using only projected measurements significantly reduces the distributional gap.*

Table 4: Comparison of InD, OOD, and Adapted models for image reconstruction using DPS, for single-coil MRI reconstruction with for acceleration ratio $R = 4$ and different measurement noise.

| **Method** | $R = 4 \quad \sigma_z = 0.00$ | | $R = 4 \quad \sigma_z = 0.01$ | |
| --- | --- | --- | --- | --- |
| | PSNR ↑ | LPIPS↓ | PSNR↑ | LPIPS↓ |
| Prostate | 24.15 | 0.3223 | 23.89 | 0.3268 |
| Knee | 26.51 | 0.2697 | 25.90 | 0.2774 |
| Brain | 27.92 | 0.2159 | 27.42 | 0.2234 |
| Adapt64 (Prostate) | 25.17 | 0.3071 | 24.80 | 0.3089 |

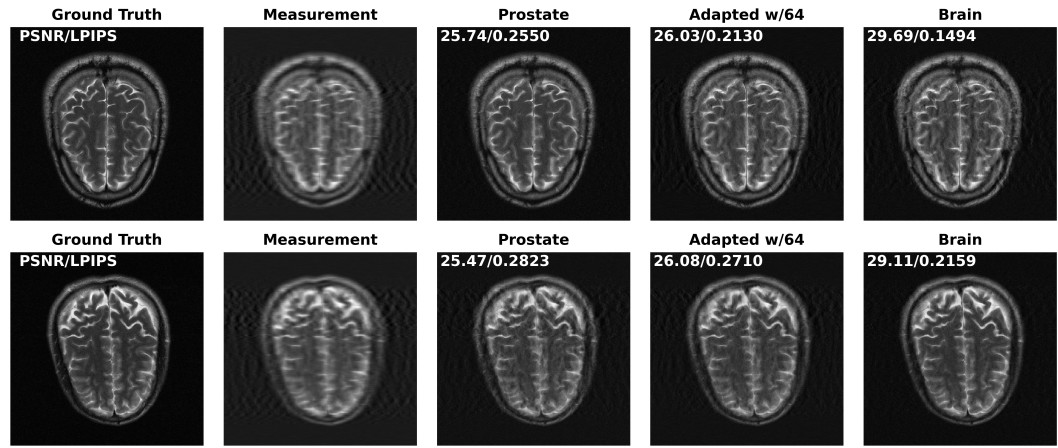

Figure 9: *Visual comparison of single-coil MRI reconstruction using DPS (Chung et al., 2023a) on a Brain MRI slice with acceleration ratio R = 4 and no measurement noise. Note the performance gap between the InD and OOD models, and the improvement achieved by adapting the OOD models using only corrupted measurements.*

Table 5 reports the KL divergence between Brain (InD) and Prostate (OOD) MRI distributions under varying acceleration rates $R \in \{4, 6, 8\}$ and measurement noise levels $\sigma_z \in \{0.0, 0.1, 0.2\}$. The most right column shows the KL divergence in the image domain. Across all settings, the measurement-domain KL estimates remain stable and closely match the image-domain value, demonstrating the robustness of our metric to both subsampling and high levels of measurement noise.

Table 5: KL divergence between Brain (InD) and Prostate (OOD) as a function of MRI acceleration rate $R$ and measurement noise level $\sigma_z$. Note the robustness of the metric to measurement noise.

| $R$ \ $\sigma_z$ | 0.0 | 0.1 | 0.2 | $D_{\mathrm{KL}}$ (Img) |
|---|---|---|---|---|
| 4 | 2.51875 | 2.53045 | 2.53055 | 2.50662 |
| 6 | 2.51844 | 2.53003 | 2.53027 | 2.50662 |
| 8 | 2.51821 | 2.52980 | 2.53002 | 2.50662 |

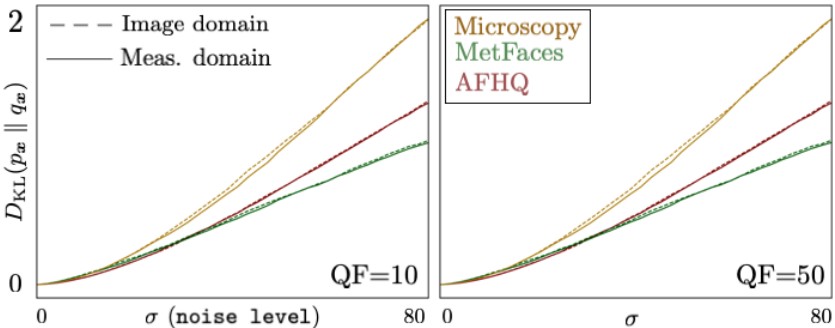

Figure 10: *Measurement-domain versus image-domain KL divergence under JPEG compression. Although JPEG artifacts violate the assumptions of Theorem 1, the measurement-domain KL remains aligned with the image-domain KL, demonstrating robustness beyond the idealized theoretical conditions.*

### D.6 COMPARISON WITH DOMAIN/TEST-TIME ADAPTATION (HU ET AL., 2025)

To contextualize our approach within the broader landscape of adaptation for inverse problems, we additionally include comparisons with recent domain/test-time adaptation methods for diffusion-based

inverse solvers, including the patch-based refinement approach of Hu et al. (2025). These methods refine the score network at inference time using a self-supervised loss that enforces measurement consistency, allowing the prior to adjust toward the test distribution. We implemented these adaptation modules using self-supervised loss $L(\theta) = \|\boldsymbol{y} - \boldsymbol{A}\widehat{\mathsf{D}}_\theta(\boldsymbol{x}_t|\boldsymbol{y})\|_2^2$, where the DPS is used as the diffusion inverse solver. Every $K = 20$ steps in the EDM sampler, the model's weight are updated using the self-supervised loss. The results are included in Table 6.

Table 6: *Comparison of InD, OOD, and Adapted models for image reconstruction using DPS (Chung et al., 2023a), for inpainting with different inpainting masks and measurement noise.* **Best** and **second best** are shown.

| Method | $p = 0.8$ $\sigma_{\boldsymbol{z}} = 0.01$ | | $p = 0.9$ $\sigma_{\boldsymbol{z}} = 0.00$ | |
|---|---|---|---|---|
| | PSNR ↑ | LPIPS↓ | PSNR↑ | LPIPS↓ |
| Microscopy | 21.68 | 0.1466 | 25.14 | 0.0707 |
| MetFaces | 25.49 | 0.0766 | 29.60 | 0.0342 |
| AFHQ | 25.84 | 0.0614 | 30.02 | 0.0246 |
| FFHQ | **28.36** | **0.0322** | **33.24** | **0.0113** |
| Adapt64 (AFHQ) | 26.14 | 0.0530 | 30.23 | 0.0208 |
| Adapt128 (AFHQ) | **26.52** | **0.0465** | **30.37** | **0.0187** |
| TTAdapt (Hu et al., 2025) | 26.39 | 0.0676 | 30.18 | 0.0246 |

## D.7 EVALUATING THE EFFECTIVE OF FIXED MRI SUBSAMPLING MASK MASK

To assess how critical Assumption 1 (randomized operators with full-span coverage) is in practice, we conducted an additional study focusing on fixed and highly structured operators, such as the Cartesian undersampling masks used in real MRI pipelines. In this setting, the forward operator no longer varies across measurements and hence does not satisfy the randomness or span conditions required by our theory. Nevertheless, our experiments show that the proposed measurement-domain KL estimator remains stable and informative even under these structured operators. Specifically, when we fix the MRI mask and evaluate the KL curve using only this single operator, the resulting measurement-domain KL continues to track the image-domain KL closely and preserves the correct model ranking. This indicates that, although our formal guarantees rely on randomized operators, the metric remains practically feasible in realistic imaging pipelines where only a single acquisition model is available. Importantly, this experiment demonstrates that the metric does not collapse under fixed operators: the residuals still reflect the mismatch between priors, and the KL curves remain smooth and monotonic, enabling reliable model comparison and adaptation guidance. Figure 11 demonstrates the results of distribution shift measurement using the proposed metric. Figure 13 presents visual comparisons. The KL divergence estimated using image domain KL and our proposed for adaptation strategy using the fixed mask is shown in Figure 12.

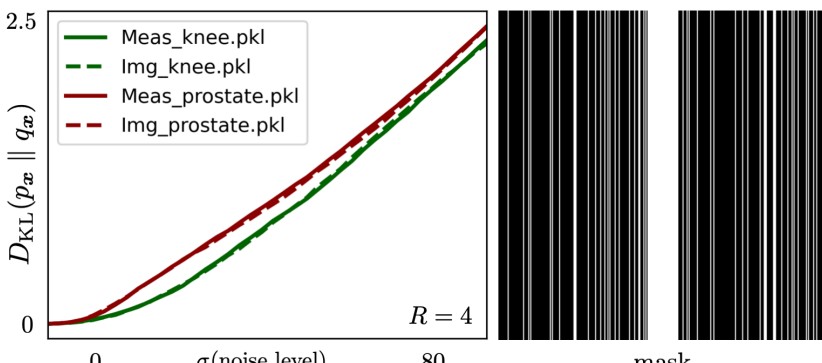

Figure 11: *Comparison of the distribution shift (dashed lines), computed using clean images, and our proposed measurement-domain KL metric (solid lines) between an InD model trained on Brain and OOD models trained on Knee and Prostate MRI slices from fastMRI dataset. Results are shown under MRI acceleration rate $R = 4$ with a fixed mask. The vertical axis shows $D_{\mathrm{KL}}$, evaluated as the integrand in Eq. (9) and Eq. (4) up to diffusion noise level $\sigma$. Right: The fixed mask used. Note that while the measurement operator does not satisfy the theoretical assumption, the metric for measuring distribution shift is shown to be practical.*

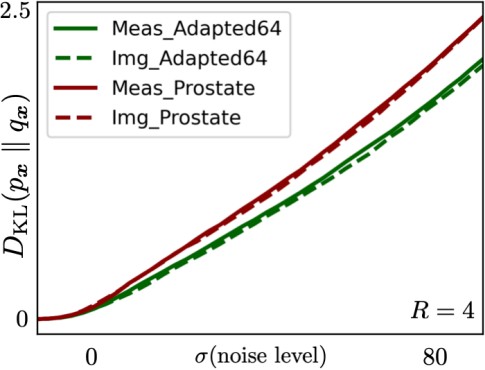

Figure 12: *Comparison of the distribution shift (dashed lines), computed using clean images, and our proposed measurement-domain KL metric (solid lines) between an InD model trained on Brain and OOD models trained on Prostate and adapted models MRI slices from fastMRI dataset. Results are shown under MRI acceleration rate $R = 4$ with a fixed mask. Note that while the measurement operator does not satisfy the theoretical assumption, the adaptation using the proposed method is shown to be practical.*

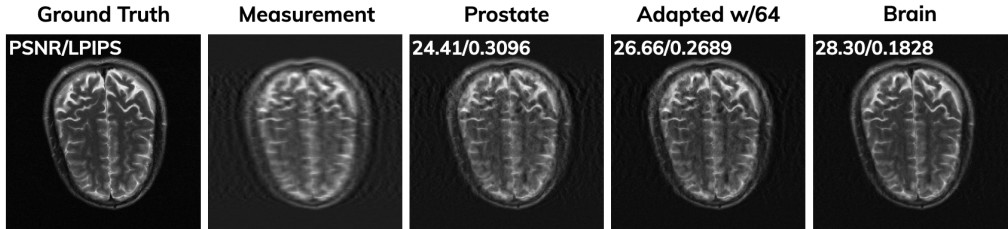

Figure 13: *Visual comparison of single-coil MRI reconstruction using DPS (Chung et al., 2023a) on a Brain fastMRI slice with acceleration ratio $R = 4$ and no measurement noise **with fixed MRI mask**. Note the performance gap between the InD (Brain) and OOD models (Prostate), and the improvement achieved by adapting (Adapted64) the OOD models using fixed measurement operator.*

## D.8 EVALUATING DISTRIBUTION SHIFT UNDER SUBTLE CONTRAST VARIATIONS-T1 VS.T2 MRI

To further examine the sensitivity of the proposed metric, we conducted an experiment targeting subtle distribution shifts—cases where the underlying anatomy is unchanged but the image contrast differs, such as between T1-weighted and T2-weighted MRI sequences. Unlike the large semantic shifts considered in our main experiments, brain vs. knee and prostate MRI, contrast differences represent a much finer-grained shift that is known to challenge OOD detectors. In this setup, we trained an InD diffusion prior on T2-weighted brain slices and used T1-weighted slices as the OOD distribution. We then computed both the measurement-domain KL curves and the image-domain KL curves over the diffusion noise levels. Figure 14 illustrates the results of KL divergence measurement. Note that the measurement-domain KL follows the image-domain KL, even with subtle distribution shift. The metric also preserves the correct ordering when compared to knee prior, demonstrating that it can distinguish between subtle and large distribution shifts. This experiment confirms that the proposed estimator is not limited to coarse anatomical differences but can reliably quantify delicate contrast-induced shifts common in clinical MRI pipelines. Figure 14 demonstrates the KL divergence estimation for Adapted prior and Figure 15 illustrates visual comparison.

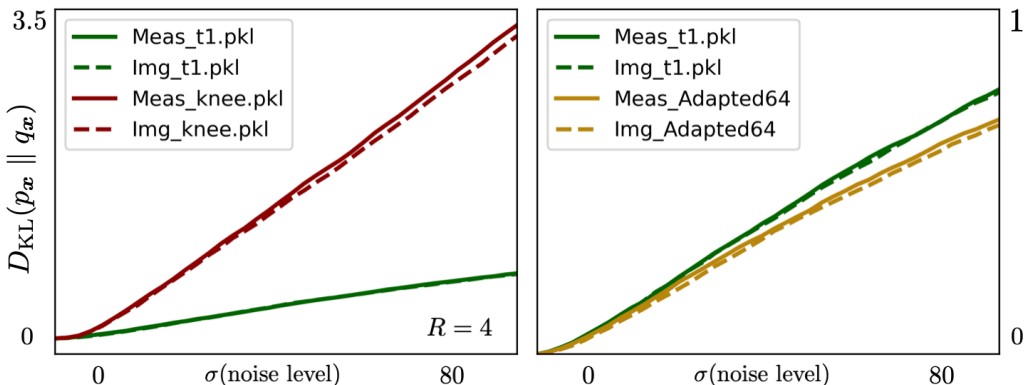

Figure 14: *Left: comparison of img-domain and our proposed measurement domain shift distribution estimation between InD (Brain T2) and OOD models (Brain T1 and Knee). Right: evaluating the effect of adaptation on the KL divergence.*

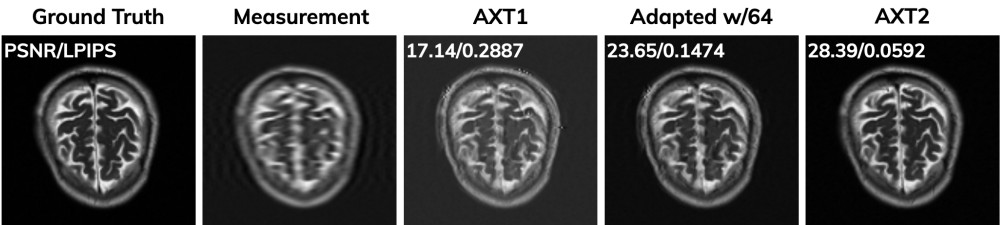

Figure 15: *Visual comparison of single-coil MRI reconstruction using DPS (Chung et al., 2023a) on a Brain T2 MRI slice with acceleration ratio $R = 4$ and no measurement noise. Note the performance gap between the InD (T2) and OOD models (T1), and the improvement achieved by adapting (Adapted64) the OOD models using only corrupted measurements.*

## D.9 ADAPTING METFACES OOD MODEL TO FFHQ

To confirm that our adaptation framework is not specific to a particular OOD prior, we repeated the measurement-only adaptation experiment using the MetFaces diffusion model as the OOD prior and FFHQ as the target distribution. The procedure mirrors the AFHQ→FFHQ adaptation pipeline described in Sections 4.2 and 4.3. The results demonstrate that adaptation also benefits

the MetFaces prior. First, the measurement-domain KL curves show a consistent downward shift for both Adapt64 and Adapt128 relative to the unadapted MetFaces model, indicating reduced prior mismatch. Second, reconstruction experiments using DPS reveal that the adapted MetFaces models achieve higher PSNR and lower LPIPS than the unadapted prior, shrinking the gap toward the FFHQ InD baseline. As expected, Adapt128 provides the largest improvement due to the increased number of measurement samples used during fine-tuning. These findings confirm that the proposed measurement-only adaptation is model-agnostic: it effectively improves any OOD diffusion prior—whether AFHQ or MetFaces—by aligning its score function with the target distribution using only corrupted measurements. The adaptation KL divergence can be found in Figure 16. Quantitative results are included in Table 7 and illustration of visual performance can be found in Figure 17.

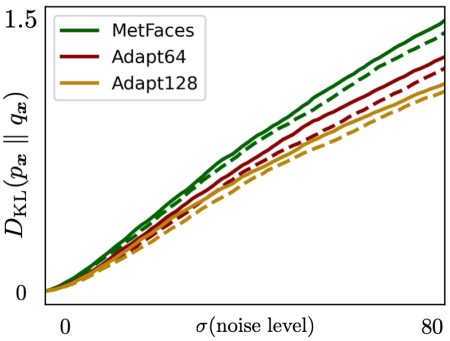

Figure 16: *KL divergence between FFHQ and MetFaces, along with adapted models using 64 and 128 projected measurements. Values are computed in the image domain (dashed) and measurement domain (solid) under inpainting with $p = 0.8$. Adaptation using only projected measurements reduces the distributional gap.*

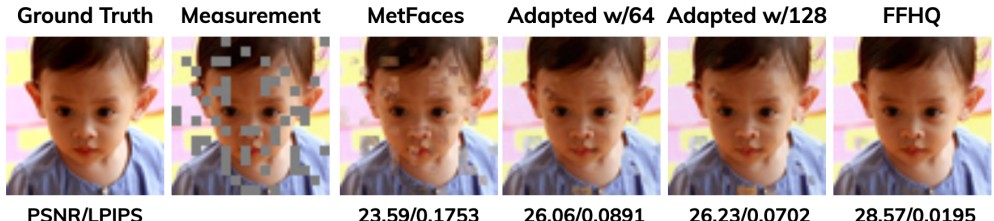

Figure 17: *Visual comparison of inpainting results (DPS (Chung et al., 2023a)) on an FFHQ image with mask rate $p = 0.8$ and measurement noise level $\sigma = 0.01$. Note the performance gap between the InD and OOD models, and the improvement achieved by adapting the OOD models using only corrupted measurements.*

Table 7: *Comparison of InD (FFHQ), OOD (MetFaces), and Adapted models for image reconstruction using DPS (Chung et al., 2023a), for inpainting with different inpainting masks and measurement noise.*

| Method | $p = 0.8$ $\quad \sigma_z = 0.01$ | | $p = 0.9$ $\quad \sigma_z = 0.00$ | |
| --- | --- | --- | --- | --- |
| | PSNR $\uparrow$ | LPIPS$\downarrow$ | PSNR$\uparrow$ | LPIPS$\downarrow$ |
| MetFaces | 25.49 | 0.0766 | 29.60 | 0.0342 |
| FFHQ | 28.36 | 0.0322 | 33.24 | 0.0113 |
| Adapt64 (MetFaces) | 26.39 | 0.0631 | 30.01 | 0.0271 |
| Adapt128 (MetFaces) | 26.78 | 0.0591 | 30.19 | 0.0259 |

## D.10 Evaluating the effect of larger measurement set in adaptation and adaption using images

Table 8 evaluates how adaptation performance scales as more measurement samples become available. Starting from the AFHQ prior, we adapt the model using 64, 128, and 256 projected measurements under two inpainting settings. As expected, increasing the number of measurements yields steady improvements in both PSNR and LPIPS, with Adapt256 consistently outperforming Adapt64 and Adapt128. These results confirm that the proposed measurement-domain adaptation benefits from additional data and continues to shrink the gap between the OOD prior and the InD diffusion model. Figure 18 illustrates the KL divergence estimation with more measurements.

Table 8: *Comparison of InD, OOD, and Adapted models for image reconstruction using DPS (Chung et al., 2023a), for inpainting with different inpainting masks and measurement noise.* **Best** and **second best** *are shown.*

| Method | $p = 0.8$ $\sigma_z = 0.01$ | | $p = 0.9$ $\sigma_z = 0.00$ | |
|---|---|---|---|---|
| | PSNR ↑ | LPIPS ↓ | PSNR ↑ | LPIPS ↓ |
| AFHQ | 25.84 | 0.0614 | 30.02 | 0.0246 |
| FFHQ | **28.36** | **0.0322** | **33.24** | **0.0113** |
| Adapt64 (AFHQ) | 26.14 | 0.0530 | 30.23 | 0.0208 |
| Adapt128 (AFHQ) | 26.52 | **0.0465** | 30.37 | **0.0187** |
| Adapt256 (AFHQ) | **26.85** | 0.0540 | **30.72** | 0.0236 |

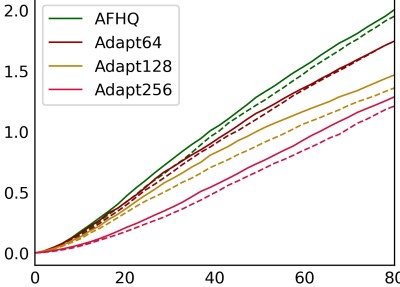

Figure 18: *KL divergence between FFHQ and AFHQ, along with adapted models using 64, 128, and 256 projected measurements. Values are computed in the image domain (dashed) and measurement domain (solid) under inpainting with $p = 0.8$. Adaptation using only projected measurements significantly reduces the gap.*

Table 9 compares measurement-only adaptation to an image-based adaptation baseline using the same number of samples. The image-based variant serves as an upper bound since it has access to clean, fully observed images. As anticipated, image-based adaptation (Adapt64/Adapt128 (img)) achieves stronger reconstruction performance. However, the measurement-only versions (Adapt64/Adapt128) is able to boost performance, despite using only corrupted measurements. This demonstrates that the proposed measurement-domain objective provides an effective and practical alternative when clean images are unavailable.

Table 9: *Comparison of InD, OOD, and Adapted models for image reconstruction using DPS (Chung et al., 2023a), for inpainting with different inpainting masks and measurement noise. Adaptation with images are also added for a upperbound on performance.*

| Method | $p = 0.8$ $\sigma_z = 0.01$ | | $p = 0.9$ $\sigma_z = 0.00$ | |
|---|---|---|---|---|
| | PSNR ↑ | LPIPS↓ | PSNR↑ | LPIPS↓ |
| AFHQ | 25.84 | 0.0614 | 30.02 | 0.0246 |
| FFHQ | 28.36 | 0.0322 | 33.24 | 0.0113 |
| Adapt64 (AFHQ) | 26.14 | 0.0530 | 30.23 | 0.0208 |
| Adapt128 (AFHQ) | 26.52 | 0.0465 | 30.37 | 0.0187 |
| Adapt64 (img) | 27.01 | 0.0543 | 30.26 | 0.0235 |
| Adapt128 (img) | 27.55 | 0.0482 | 31.03 | 0.0206 |

