# OpenReview forum: "Measuring  Distribution Shifts in Inverse Problems without clean data"
_ICLR.cc/2026/Conference — Submitted to ICLR 2026_

### Official Review · Reviewer_hv1x · 2025-10-27

**Soundness:** 3
**Presentation:** 3
**Contribution:** 2
**Rating:** 6
**Confidence:** 3

**Summary:**

The author deal with the important problem of measuring distribution shift using only measurements. They use their approach for model selection, estimation of the KL divergence and for fine-tuning / adaptation.

**Strengths:**

- The paper is well-written
- The main result in Theorem 1 is interesting and also holds in the numerical experiments
- I think in particular the experiments on the effect of the model selection and adaptation on downstream tasks (page 7, starting line 365) are interesting and important

**Weaknesses:**

See questions.



I have some minor issues with the formating:
- Figure 1: It is hard to see the difference of the plot for "image domain" and "meas. domaine" without zooming in on the PDF. In a printed version it is even harder to tell apart.
- Figure 1 is on page 3, but only discussed on page 6
- Figure 3 is discussed on page 6 (line 309), but is only displayed on page 8 (after Figure 6)
- Page 7, line 377 the subsection title "4.4 Ablation Studies" it at the end of the page, without any text below it

Other minor points:
- In Assumption 2 you use $\hat{D}$, but $\hat{D}$ is only defined  later in Theorem 1
- page 7, line 348 missing space between "OOD denoiser" and $\hat{D}$

**Questions:**

- You write that your approach even works for JPEG compression, a setting not supported by the theory. What are inverse problems where your method would not work? Do you have some neccessary conditions on a problem?
- In Figure 2: Is this also the integrand of Eq. (4) and Eq. (9) up to $\sigma$?
- In Table 1 we see that MetFaces is closer to FFHQ than AFHQ. So, using your model selection criterion I would choose MetFaces. However, in Table 2 the image reconstruction results for MetFaces are worse than for AFHQ. Why?
- What is the difference between your adaptation loss in Equation (10) and the objective in Ambient Diffusion [1] or [2]
- Why do you choose AFHQ in the adaptation in 4.3. and not MetFaces (MetFaces was closer to FFHQ according to Table 1)?

[1] Daras et al. "Ambient Diffusion: Learning Clean Distributions from Corrupted Data" (2023)

[2] Kawar et al. "GSURE-Based Diffusion Model Training with Corrupted Data" (2024)

---

> ### Author Response · Authors · 2025-11-24
>
> We thank the reviewer for taking the time to provide feedback. Below we address each comment point-by-point.
>
> >The reviewer notes multiple minor formatting issues.
>
> All formatting corrections have been made in the revised manuscript.
>
> >You write that your approach even works for JPEG compression, a setting not supported by the theory. What are inverse problems where your method would not work? Do you have some neccessary conditions on a problem?
>
> Our positive results on JPEG are purely empirical—JPEG violates the assumptions of our theory, yet still preserves enough discriminative structure for the KL estimator to function. In general, our method succeeds when the measurement operator retains the distinguishing features of the underlying image distribution. Conversely, it will fail in settings where measurements collapse these features—for example, operators that produce near-identical outputs across samples, severe blurring that destroys semantic content, or degenerate pushforward mappings with identical projections. We added this clarification to the revision.
>
>
> >In Figure 2: Is this also the integrand of Eq. (4) and Eq. (9) up to $\sigma$?
>
> Yes, that is correct.
>
> >In Table 1 we see that MetFaces is closer to FFHQ than AFHQ. So, using your model selection criterion I would choose MetFaces. However, in Table 2 the image reconstruction results for MetFaces are worse than for AFHQ. Why?
>
> Although MetFaces is closer to FFHQ in distribution, it performs worse in reconstruction because its diffusion prior is trained on a much smaller dataset—1336 images versus 15000 for AFHQ, leading to a poorer prior for DPS. To isolate the effect of dataset size, we trained an AFHQ-based diffusion model using only 1,336 randomly selected images (matching the size of MetFaces) and reevaluated its reconstruction performance; the results are included in the following table and confirm that the performance gap is largely due to the smaller training set.
>
> **Table: Effect of Dataset Size on Reconstruction Using DPS**
>
> | Method          | PSNR ($p = 0.8, σ_z = 0.01$) | LPIPS | PSNR ($p = 0.9, σ_z = 0.00$) | LPIPS |
> |-----------------|----------------------------|--------|------------------------------|--------|
> | AFHQ (15k)      | 25.84                     | 0.0614 | 30.02                        | 0.0246 |
> | AFHQ (1336)     | 25.21                     | 0.0836 | 27.57                        | 0.0431 |
> | MetFaces (1336) | 25.49                     | 0.0766 | 29.60                        | 0.0342 |
>
>
> > What is the difference between your adaptation loss in Equation (10) and the objective in Ambient Diffusion [1] or [2]
>
> Our adaptation loss (Eq. 10) is mathematically different from both GSURE and Ambient Diffusion and also serves a different purpose. The GSURE ideal loss (Eq. 9),
>
> $L_{\text{GSURE}} =\sum_{t=1}^T \gamma_t \mathbb{E} \Big[\Vert W P( f_\theta^{(t)}(\bar{x}_t)-\bar{x})\Vert \Big],$
>
> aims to match the model output $f$ to the clean latent variable $\bar{x}$ on the observed coordinates defined by P, and in practice it adds a divergence term to form an unbiased GSURE estimator. In contrast, our loss aligns $\hat{D} (V\bar y_\sigma)$ with $V\bar y$, a measurement‐domain projection, and never supervises on $\bar{x}$. GSURE (Eq. 9/10) is therefore a GSURE-corrected surrogate for the supervised diffusion loss, while our Eq. 10 is a KL-motivated alignment loss that reduces the discrepancy between the InD and OOD denoiser residuals in the measurement domain.
>
> Ambient Diffusion is also fundamentally different: it introduces an additional corruption $\tilde A$, feeds the doubly-corrupted signal $\tilde A x_t$ into the network, and measures the error against $A x_0$.
>
> In summary, GSURE and Ambient Diffusion both aim to learn the clean score function directly from corrupted measurements. Our adaptation loss does not attempt to learn the clean score; instead, it fine-tunes a pretrained OOD model so that its score function aligns with the InD score on projected measurements, as motivated by Theorem 1.
>
>
> > Why do you choose AFHQ in the adaptation in 4.3. and not MetFaces (MetFaces was closer to FFHQ according to Table 1)?
>
> We selected AFHQ for adaptation simply as a design choice, and our method is not specific to that model. To verify this, we also adapted the MetFaces prior using 64 and 128 projected measurements and reported the corresponding results in Section D.9 of the supplement. These results confirm that the adaptation procedure works similarly for MetFaces as well.
>
> **Table: Adaptation Results for MetFaces Prior**
>
> | Method     | PSNR ($p = 0.8, σ_z = 0.01$) | LPIPS | PSNR ($p = 0.9, σ_z = 0.00$) | LPIPS |
> |------------|----------------------------|--------|------------------------------|--------|
> | MetFaces      | 25.49 | 0.0766 | 29.60 | 0.0342 |
> | Adapt64       | 26.39 | 0.0631 | 30.01 | 0.0271 |
> | Adapt128      | 26.78 | 0.0591 | 30.19 | 0.0259 |
> | FFHQ (InD)    | 28.36 | 0.0322 | 33.24 | 0.0113 |

---

> ### Author Response · Authors · 2025-11-26
>
> Dear Reviewer hv1x,
>
> Thank you again for reviewing our paper. We did our best to address your comments and would appreciate any post-rebuttal feedback. Let us know if any additional details would be helpful in supporting a more positive assessment of our work.

---

> > ### Comment · Reviewer_hv1x · 2025-11-27
> >
> > Thank you for the rebuttal and the additional clarifications. I am sorry for my late response, but I will note that the ICLR guidelines suggested to upload the rebuttal by November 20.
> >
> > I acknowledge the updates to the manuscript. The new experiments on dataset size are particularly helpful.
> >
> > **MetFaces vs. AFHQ (1336)**
> >
> > Given your model-selection score, which dataset would you select, MetFaces (1336) or AFHQ (1336)?
> >
> > **Adaptation Loss**
> >
> > Thanks for the clarification regarding Ambient Diffusion and GSURE. I have a few follow-up questions:
> >
> > 1. How do you compute $W^2 = \mathbb{E}[H^\dagger H]$ in your experiments?
> >
> > 2. Does the adaptation loss in Equation (10) remain valid when the measurements are noisy, i.e., $y = Px + n$, where $n$ is measurement noise rather than diffusion-model noise?
> >
> > **Other Forward Operators**
> >
> > A bit of a broader question: the framework depends on access to the SVD of $H$, which restricts it to linear forward operators and, in practice, to those with tractable SVDs. Do you see any way of extending this approach to nonlinear inverse problems?

---

> ### Author Response · Authors · 2025-12-02
>
> > MetFaces vs. AFHQ (1336). Given your model-selection score, which dataset would you select?
>
> Prompted by your comment, we provide a comparison of the selection metric for **MetFaces (1336)**, **AFHQ (1336)**, and **AFHQ (15k)** (complementary to Table 1).
>
> #### **Selection Metric Table**
>
> | p   | InD (FFHQ) | OOD MetFaces (1336) | OOD AFHQ (15k) | OOD AFHQ (1336) |
> |-----|------------|----------------------|------------------|------------------|
> | 0.9 | 1.677      | 3.561                | 4.009            | 4.444            |
> | 0.8 | 1.805      | 3.651                | 4.076            | 4.497            |
> | 0.7 | 1.929      | 3.713                | 4.154            | 4.553            |
> | 0.5 | 2.067      | 3.730                | 4.186            | 4.658            |
>
> Note that the metric still chooses MetFaces as the best match for the measurements.
>
> > How do you compute $W\^2 \= \\mathbb\{E\}\[H\^\\dagger H\] $ in your experiments?
>
> In inpainting, each $H$ is a diagonal Bernoulli mask, so $H^\dagger H = P$ and the expectation is $W^2=\mathbb{E}[P]=\pi I$, with $\pi$ the keep probability. For MRI, $H = I \Sigma F$ with unitary F; in the SVD/measurement coordinates $H^\dagger H$ becomes the diagonal projector $P=\mathrm{diag}(\sigma_k).$ Thus $W^2=\mathbb{E}[H^\dagger H]=\mathrm{diag}(\pi_k)$, the k-space sampling probabilities, which we precompute from the mask distribution.
>
> > Does the adaptation loss in Eq. (10) remain valid under noisy measurements $ y \= Hx \+ z $?
>
> Yes—Eq. (10) remains fully valid under noisy measurements. As clarified at the end of Section B of the Supplement, our derivation is carried out under the general model $y = Hx + z$, so measurement noise is already accounted for. Moreover, Tables 2 and 3 empirically verify that both the model-selection score and the adaptation loss remain stable across all tested noise levels.
>
> > A bit of a broader question: the framework depends on access to the SVD of , which restricts it to linear forward operators and, in practice, to those with tractable SVDs. Do you see any way of extending this approach to nonlinear inverse problems?
>
> For nonlinear forward operators, we agree that the SVD-based analysis used in our linear setting is no longer applicable. A more principled extension would instead work with the pushforward distribution of the images under the nonlinear map. The KL formulation only requires access to the score of the image distribution; for a nonlinear measurement operators, one could consider estimating the score of the induced measurement distribution $p_y$ and relate it to the image score via pushforward score identities (for example by transporting $\nabla_y \log p_y$ back through the Jacobian of $f$ when feasible). This would provide a pathway to generalize our KL-based shift measure without relying on an SVD. Developing such a pushforward-score framework for fully nonlinear operators is nontrivial and beyond the scope of this work, but it represents a promising direction.

---

### Official Review · Reviewer_AU8b · 2025-10-31

**Soundness:** 3
**Presentation:** 3
**Contribution:** 3
**Rating:** 6
**Confidence:** 3

**Summary:**

This paper introduces a method for detecting distribution shifts between measurements and priors to (1) select the best prior for regularization (2) mitigate distribution shift effects at inference time when using an OOD prior. The authors make an important point that often in inverse problem settings we do not have access to clean images to measure distribution shifts with the prior and thus there is a need for a unsupervised approach. Experimental results show good alignment between their method and image-based estimates for OOD detection. Additionally, the authors show that using an augmented loss to correct for out of distribution samples they can adapt OOD models with few samples and show improvement for said models on downstream recovery.

**Strengths:**

This paper solves an interesting problem that is important for deploying pre-trained generative models to image recovery in potentially new settings. As far as I know this is the first works to look into this for inverse problems in the self-supervised setting. This setting is crucial because adapting generative priors to new scientific data will likely require pre-trained models that havent been explicitly trained on the target distribution due to data scarcity. The paper has nice experiments showing the ability of their method to reliably detect OOD models from partial measurments. Additionally, the authors show encouraging results of using their technique to adapt their OOD models to the target distribution using only a limited number of samples which is really great to see.

**Weaknesses:**

The results are convincing for the most part, however, it would be nice to potentially see a few more experiments with more samples of the measurement data to get a better idea of how the method scales with more measurement data when adapting OOD models to the target distribution. Additionally, if it’s possible to compare their measurement only adaption approach to image-based adaptation approaches with with the same number of samples that would be a helpful baseline which can serve as the upper bound on performance for their measurement adaption otherwise its a bit more difficult to appreciate the performance gains they are getting.

**Questions:**

1. how does the method scale in performance with a higher # of measurement examples?
2. how does the method compare to image based adaption techniques with the same number of samples?

---

> ### Author Response · Authors · 2025-11-24
>
> We thank the reviewer for taking the time to provide feedback. Below we address each comment point-by-point.
>
> >The results are convincing for the most part, however, it would be nice to potentially see a few more experiments with more samples of the measurement data to get a better idea of how the method scales with more measurement data when adapting OOD models to the target distribution.
>
> >how does the method scale in performance with a higher # of measurement examples?
>
> Prompted by this comment, we added new experiments using **256 measurement samples** for adaptation.  As reported in **Section D.10 (supplement)**, these results show that performance improves steadily as the number of measurement samples increases.
>
> **Table: Scaling of Measurement-Only Adaptation with More Samples**
>
> | Method    | PSNR ($p = 0.8, σ_z = 0.01$) | LPIPS | PSNR ($p = 0.9, σ_z = 0.00$) | LPIPS |
> |-----------|----------------------------|--------|------------------------------|--------|
> | AFHQ      | 25.84                     | 0.0614 | 30.02                        | 0.0246 |
> | FFHQ      | 28.36                     | 0.0322 | 33.24                        | 0.0113 |
> | Adapt64   | 26.14                     | 0.0530 | 30.23                        | 0.0208 |
> | Adapt128  | 26.52                     | 0.0465 | 30.37                        | 0.0187 |
> | **Adapt256** | **26.85**             | **0.0540** | **30.72**                    | **0.0236** |
>
> These results confirm that measurement-domain adaptation **scales smoothly with more samples**.
>
>
> > Additionally, if it’s possible to compare their measurement only adaption approach to image-based adaptation approaches with with the same number of samples that would be a helpful baseline which can serve as the upper bound on performance for their measurement adaption otherwise its a bit more difficult to appreciate the performance gains they are getting.
>
> >how does the method compare to image based adaption techniques with the same number of samples?
>
> Prompted by this comment, we added an **image-based adaptation baseline** using the same number of samples (64 and 128).  These results are included in **Section D.10** and provide a meaningful upper bound for comparison.
>
> **Table: Comparison of Measurement-Only vs Image-Based Adaptation**
>
> | Method        | PSNR ($p = 0.8, σ_z = 0.01$) | LPIPS | PSNR ($p = 0.9, σ_z = 0.00$) | LPIPS |
> |---------------|----------------------------|--------|------------------------------|--------|
> | AFHQ          | 25.84                     | 0.0614 | 30.02                        | 0.0246 |
> | FFHQ          | 28.36                     | 0.0322 | 33.24                        | 0.0113 |
> | Adapt64       | 26.14                     | 0.0530 | 30.23                        | 0.0208 |
> | Adapt128      | 26.52                     | 0.0465 | 30.37                        | 0.0187 |
> | **Adapt64_img**  | **27.01**             | **0.0543** | **30.26**                    | **0.0235** |
> | **Adapt128_img** | **27.55**             | **0.0482** | **31.03**                    | **0.0206** |

---

> ### Author Response · Authors · 2025-11-26
>
> Dear Reviewer AU8b,
>
> Thank you again for reviewing our paper. We did our best to address your comments and would appreciate any post-rebuttal feedback. Let us know if any additional details would be helpful in supporting a more positive assessment of our work.

---

### Official Review · Reviewer_wiAN · 2025-10-31

**Soundness:** 2
**Presentation:** 2
**Contribution:** 2
**Rating:** 4
**Confidence:** 3

**Summary:**

This paper introduces an unsupervised framework to measure distribution shifts in inverse problems using only corrupted measurements, removing the need for clean test images. It reformulates the KLD between in-distribution and out-of-distribution data through diffusion model score functions evaluated in the measurement domain. The derived estimator links shift magnitude to denoiser residuals across noise levels, enabling model selection, divergence estimation, and adaptation that aligns out-of-distribution scores with measurement data. Experiments on image inpainting and magnetic resonance imaging show strong correspondence between image- and measurement-domain estimates and improved reconstruction after adaptation.

**Strengths:**

The paper presents an interesting and timely idea by proposing a framework to quantify distribution shifts in inverse problems without access to clean data. The formulation is conceptually clear, and the paper is generally well written and structured, with good alignment between theory and experiments. The method shows potential practical value by enabling model selection and adaptation using only measurement data. However, while these aspects are promising, the overall contribution remains somewhat limited in scope and especially experimental depth.

**Weaknesses:**

My main concern lies in the experimental setup, which appears overly simplified and somewhat artificial. The chosen tasks, such as low-resolution inpainting and small-scale fastMRI tests, do not capture the real challenges of distribution shift in medical imaging. The out-of-distribution settings, based on different anatomical regions or synthetic corruptions, are relatively easy to separate and may overstate the method’s performance. A more convincing validation would use realistic shifts, for example differences in scanner field strength (1.5 T versus 3 T MRI), acquisition protocols, or vendor-specific pipelines, where data differ in subtle but meaningful ways and clean reference images are not available. This would better demonstrate the method’s practical value and robustness.

Another weakness is the strong reliance on assumptions that rarely hold in practice, such as randomized measurement operators and independence between measurements and denoiser residuals. These conditions are violated in realistic setups like fixed MRI masks, making the theoretical guarantees and practical reliability of the method uncertain.

**Questions:**

1. The experiments appear simplified and artificial. How would the proposed framework perform under more realistic domain shifts, such as differences in scanner field strength (e.g., 1.5 T vs 3 T MRI), acquisition protocols, or vendor-specific reconstruction pipelines?

2. Given that the out-of-distribution scenarios used in the paper (different anatomies or synthetic corruptions) are relatively easy to distinguish, can the authors provide evidence that their metric remains reliable when the domain shift is subtle but clinically meaningful?

3. The theoretical analysis assumes randomized measurement operators whose span covers the full signal space. How critical is this assumption in practice, and what happens if one uses fixed or structured operators as in real MRI or CT acquisition?

4. Since the proposed metric depends on expectations over many random operators, how feasible is this in realistic imaging pipelines where only a single, fixed acquisition model is available?

---

> ### Author Response · Authors · 2025-11-24
>
> We thank the reviewer for taking the time to provide feedback. Below we address each comment point-by-point.
>
> > My main concern lies in the experimental setup, which appears overly simplified and somewhat artificial. The chosen tasks, such as low-resolution inpainting and small-scale fastMRI tests, do not capture the real challenges of distribution shift in medical imaging. The out-of-distribution settings, based on different anatomical regions or synthetic corruptions, are relatively easy to separate and may overstate the method’s performance. A more convincing validation would use realistic shifts, for example differences in scanner field strength (1.5 T versus 3 T MRI), acquisition protocols, or vendor-specific pipelines, where data differ in subtle but meaningful ways and clean reference images are not available. This would better demonstrate the method’s practical value and robustness.
>
>
> Prompted by your comment, we added an experiment evaluating **subtle but clinically meaningful distribution shifts**—specifically **T1 vs. T2 MRI**.  These results, included in **Section D.8 of the supplement**, confirm that the **measurement-domain KL remains sensitive even when the shift is small**.  This demonstrates that the proposed metric continues to detect shifts beyond simple anatomical or synthetic variations.
>
>
> > The experiments appear simplified and artificial. How would the proposed framework perform under more realistic domain shifts, such as differences in scanner field strength (e.g., 1.5 T vs 3 T MRI), acquisition protocols, or vendor-specific reconstruction pipelines?
>
> Our **T1 vs. T2** experiment provides a **controlled, clinically relevant surrogate** for protocol-induced shifts.  The results show that the KL-based metric remains robust under subtle domain changes, supporting its applicability to realistic medical imaging shifts.
>
>
> >Another weakness is the strong reliance on assumptions that rarely hold in practice, such as randomized measurement operators and independence between measurements and denoiser residuals. These conditions are violated in realistic setups like fixed MRI masks, making the theoretical guarantees and practical reliability of the method uncertain.
>
> >How critical are these assumptions, and what happens with fixed or structured operators such as those used in MRI or CT?
>
> >Since the metric depends on expectations over random operators, how feasible is this when only a single fixed acquisition model is available?
>
> While the  theoretical analysis relies on randomized operators, our experiments show that these conditions are **not required in practice**. Prompted by this comment, we added an experiment using a **fixed Cartesian MRI mask**, where the operator is **fully deterministic** and its span does **not** cover the full signal space.
>
> As shown in **Section D.7**, the **measurement-domain KL continues to track the image-domain KL closely**, indicating that the metric remains stable,  informative, and practical even under realistic, non-randomized acquisition settings. This suggests that the theoretical assumptions are sufficient for analysis but  not necessary for the method to work in practice.

---

> ### Author Response · Authors · 2025-11-26
>
> Dear Reviewer wiAN,
>
> Thank you again for reviewing our paper. We did our best to address your comments and would appreciate any post-rebuttal feedback. Let us know if any additional details would be helpful in supporting a more positive assessment of our work.

---

### Official Review · Reviewer_DSpX · 2025-11-01

**Soundness:** 2
**Presentation:** 2
**Contribution:** 2
**Rating:** 4
**Confidence:** 4

**Summary:**

The author proposes a method to measure distributional shift in inverse problems at test-time.

**Strengths:**

1. Use diffusion model to detect OOD problems is interesting.
2. The mathematics in this paper looks correct to me.

**Weaknesses:**

1. This paper should also compare with domain/test-time adaptation methods for inverse problems. such as [1], [2].
2. There are literatures on using diffusion models to detect OOD samples, e.g. perturbing intermediate noise or so on. Authors should mention them.[3]
3. The novelty of this work is lacking since OOD detection with diffusion model is well known, and adapting a pretrained model to OOD inverse problem is also well-known. The authors contribution in this field is obscure.


[1] Deep Diffusion Image Prior for Efficient OOD Adaptation in 3D Inverse Problems

[2] Patch-based diffusion models beat whole-image models for mismatched distribution inverse problems

[3] Denoising diffusion models for out-of-distribution detection

**Questions:**

Instead of selecting the best pretrained model, is there a way to improve the pretrained model for inverse problem solving given your OOD detection information?

---

> ### Author Response · Authors · 2025-11-24
>
> We thank the reviewer for taking the time to provide feedback. Below we address each comment point-by-point.
> >This paper should also compare with domain/test-time adaptation methods for inverse problems, such as [1], [2].
>
> Prompted by your comment, we added direct comparisons with domain/test-time adaptation [2].  These results appear in **Section D.6 (Table 6)** of the revised paper.
>
> **Table: Comparison with Test-Time / Domain Adaptation Methods**
>
> | Method    | PSNR ($p = 0.8, σ_z = 0.01$) | LPIPS | PSNR ($p = 0.9, σ_z = 0.00$) | LPIPS |
> |-----------|----------------------------|--------|------------------------------|--------|
> | AFHQ      | 25.84                     | 0.0614 | 30.02                        | 0.0246 |
> | FFHQ      | 28.36                     | 0.0322 | 33.24                        | 0.0113 |
> | Adapt64   | 26.14                     | 0.0530 | 30.23                        | 0.0208 |
> | Adapt128  | 26.52                     | 0.0465 | 30.37                        | 0.0187 |
> | Adapt [2] | 26.39                     | 0.0676 | 30.18                        | 0.0246 |
>
> > There is prior literature on using diffusion models for OOD detection (e.g., perturbing intermediate noise levels). Authors should mention these works [3].
>
> We agree and clarify that diffusion-based OOD detection is already discussed in the paper (L049–L052).  Prompted by this comment, we now make this connection explicit in **Section C (Related Work)** of the supplement.
>
> >The novelty is lacking since OOD detection with diffusion models is known, and adapting a pretrained model to OOD inverse problems is also known. The contribution is unclear.
>
> We respectfully clarify that **no prior work addresses OOD detection or shift quantification in *measurement space*.**  This distinction is crucial:  in imaging inverse problems, clean images are *not available*, and existing diffusion-based OOD methods operate **exclusively on clean images**, making them unusable in the measurement domain.
>
>
> > Instead of selecting the best pretrained model, can the pretrained model be improved using your OOD detection tools?
>
> Yes. Our method goes beyond model selection.  As shown in **Section 4.3** and **Appendix D.5**, the KL metric **directly serves as a measurement-domain loss function**, enabling improvement of an OOD prior through adaptation.
>
> [1] *Deep Diffusion Image Prior for Efficient OOD Adaptation in 3D Inverse Problems*
> [2] *Patch-based Diffusion Models Beat Whole-Image Models for Mismatched Distribution Inverse Problems*
> [3] *Denoising Diffusion Models for Out-of-Distribution Detection*

---

> ### Author Response · Authors · 2025-11-26
>
> Dear Reviewer DSpX,
>
> Thank you again for reviewing our paper. We did our best to address your comments and would appreciate any post-rebuttal feedback. Let us know if any additional details would be helpful in supporting a more positive assessment of our work.

---

### Author Response · Authors · 2025-11-24
**Summary of response**

We thank the reviewers and the area chair for taking the time to read our paper and provide valuable feedback. In response, we performed several additional experiments to address all comments:

- **Subtle clinical distribution shifts (T1 vs. T2 MRI):** Demonstrating that the measurement-domain KL remains sensitive even when shifts are small *(prompted by reviewer wiAN)*.

- **Fixed Cartesian MRI mask:**  Showing that the method remains practical and reliable even when the measurement operator is fixed  *(prompted by reviewer wiAN)*.

- **Adaptation with more measurements:**  Evaluating performance using 256 measurements and confirming improved adaptation with more data  *(prompted by reviewer AU8b)*.

- **Image-based adaptation baseline:**  Adding a matched-sample upper-bound comparison using clean-image adaptation  *(prompted by reviewer AU8b)*.

- **MetFaces adaptation experiment:** Confirming that measurement-only adaptation is model-agnostic  *(prompted by reviewer hv1x)*.

- **Comparison with test-time adaptation:**  Adding results for diffusion-based test-time refinement *(prompted by reviewer DSpX)*.

---

### Comment · Area_Chair_VTeQ · 2025-11-26

Dear Reviewers:

We kindly encourage you to take a moment to review the authors’ rebuttals and submit your feedback. Your prompt feedback is important for ensuring a thorough review. Thank you for your contributions to ICLR 2026. If you have responded to the authors' rebuttal, please feel free to ignore this message.

Thanks,
AC

---

### Author Response · Authors · 2025-12-02
**Summary of rebuttal**

Dear Area Chair,

Given the shortened discussion window, we would like to provide a brief  summary of the reviewers’ concerns, the corresponding revisions and new experiments we conducted, and how the reviewers themselves characterized the strengths of the paper.

Several reviewers describe the work as *timely, interesting,* and *well aligned with both theory and experiments.* One reviewer notes that, to the best of their knowledge, this is the **first work** to study *unsupervised distribution-shift measurement for inverse problems using only measurements and pre-trained generative priors*. Another reviewer confirms that **Theorem 1 is interesting** and that its conclusions are *borne out empirically*, indicating strong conceptual validity.

#### **Summary of Revisions and New Experiments**

- **Experiments were “simplified and somewhat artificial”; need clinically meaningful shifts (R1/wiAN).**

   Added T1 vs. T2 MRI protocol shift experiments, demonstrating sensitivity to subtle, realistic medical-domain changes (Section D.8).

- **Practicality of the method under measurement operator with fixed structure such as fixed MRI subsampling mask (wiAN).**

  Added a fixed Cartesian MRI mask experiment, showing our KL-based estimator still matches image-domain KL under a structured, single operator and adaption loss is still able to reduce KL and provide performance improvement in image reconstruction (Section D.7)

- **How adaptation scales with more measurement data (AU8b).**

  Added experiments with more measurement data for adaptation showing monotonic improvement as more measurements become available (Section D.10).

- **Need an image-based adaptation baseline to compare against and establish an upper bound (AU8b).**

  Added image-based adaptation baselines (Adapt64_img / Adapt128_img) for an upper bound on the performance (Section D.10).

- **Why AFHQ was chosen for adaptation and not MetFaces (hv1x).**

  Added MetFaces adaptation results (Adapt64/Adapt128), demonstrating that the adaptation is model agnostic (Section D.9).

- **Need clearer distinction between our adaptation loss and GSURE/Ambient Diffusion (hv1x).**

  Provided clarification that our loss is fundamentally KL-based and serves a different purpose from GSURE/Ambient diffusion.

- **Need comparison to test-time/domain adaptation and recent diffusion OOD baselines (DSpX).**

  Added comparison to a recent diffusion test-time adaptation method (patch-based) and revised related work to clarify the distinction from diffusion OOD methods that operate only on clean images (Section D.6).

We hope this summary helps clarify how each reviewer’s concerns were directly resolved with additional analyses and experiments. Thank you for considering our submission under the compressed review timeline.

Sincerely,
Authors

---

### Meta-Review · Area_Chair_h4jM · 2026-01-16

**Summary:**

The paper uses diffusion to detect OOD problems and measure distribution shifts in inverse problems using only currupted data. This is an interesting and important problem. While there is novelty, this paper has several issues, missing some related work and having a limited experimental evaluation.

**Reviewer Concerns:**

Reviewer DSpX raised valid concerns about insufficient experimental evaluation compared to prior work on OOD detection using diffusions. The authors added experiments to address these concerns but this type of significant addition to related work and experiments needs to be re-reviewed and goes beyond addressing reviewer questions.

Along the same lines, reviewer wiAN found the experimental setup limited and overly simplified. It would be interesting to see OOD problems using different analtomies or scanner field strength. The authors in the rebuttal discussed some additional experiments, but this constitutes a major revision in my opinion.

Overall I would like to encourage the authors to include all these additional experiments and ablations, improve the presentation and submit their work to the next top ML venue.

**Reviewer Scores:**

i think the paper needed significant more work which warrants another round of reviews in a new venue.

---

### Decision · Program_Chairs · 2026-01-26

Reject